# Decomposing Query-Key Feature Interactions Using Contrastive Covariances

**Andrew Lee**[1]   **Yonatan Belinkov**[2 3]   **Fernanda Viégas**[1 4]   **Martin Wattenberg**[1 4]

## Abstract

Despite the central role of attention heads in Transformers, we lack tools to understand why a model attends to a particular token. To address this, we study the query-key (QK) space – the bilinear joint embedding space between queries and keys. We present a contrastive covariance method to decompose the QK space into low-rank, human-interpretable components. It is when features in keys and queries align in these low-rank subspaces that high attention scores are produced. We first study our method both analytically and empirically in a simplified setting. We then apply our method to large language models to identify human-interpretable QK subspaces for categorical semantic features and binding features. Finally, we demonstrate how attention scores can be attributed to our identified features.

## 1. Introduction

Attention is at the heart of Transformers, yet we struggle to answer "why did the model attend to this token?".

Attention heads produce key and query vectors for each token, and their *dot product* yields an attention score. However, such scalar valued dot product conceals *how* the two tokens interact. We thus study the *QK space* – the bilinear joint embedding space between queries and keys.

Understanding the structure of QK spaces reveals how queries and keys interact. We demonstrate a simple way to decompose a QK space into interpretable low-rank features. As we will see, it is when these features in keys and queries align that leads to high attention scores.

Our method relies on the covariance of keys and queries. We define *positive* and *negative* covariance terms between

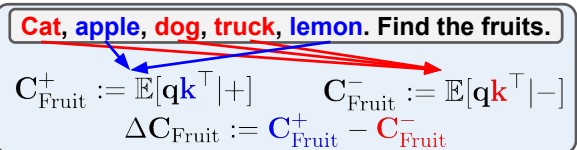

*Figure 1.* **Contrastive covariance method schema.** We define *positive* and *negative* covariance terms between queries and keys, each capturing the presence (or absence) of a feature. The resulting contrastive covariance term isolates the feature in QK space.

keys and queries, each of which correspond to the presence (or absence) of a feature of interest, while holding all other factors constant. Their difference, i.e. the *contrastive covariance*, isolates the subspace of a feature: see Figure 1. Our method allows us to 1) recover the rank of features in QK space, and 2) recover the subspaces in which features lie, in both the query and key spaces.

To show this, we design a task in which queries and keys are constructed from known latent features with ranging degrees of freedom. We show analytically that in our setting, our method recovers the correct ranks and subspaces of the latent features in query and key spaces. We then empirically verify our method by training attention heads and conducting causal interventions in our recovered QK subspaces. We also study *superposition* (Elhage et al., 2022) in QK space using our setup to study the limitations of our method.

Next, we apply our method to Llama 3.1-8B Instruct (Grattafiori et al., 2024) and Qwen 3-4B Instruct (Yang et al., 2025) to find interpretable, low-rank QK subspaces. We study two examples where attention plays a central role: categorical semantic features in Filter Heads (Sharma et al., 2025) and binding features (Gur-Arieh et al., 2025). While prior works demonstrate such mechanisms, we localize the subspaces in which they are encoded.

Finally, we show how attention logits (attention scores prior to softmax) can be attributed to the QK features that we identify. This follows naturally from the logits being linear in query space, thus decomposing the query space directly allows us to decompose the logit space. Put differently, we can identify how much each feature component contributes towards the final attention logits, but also inform on how much of the logits are left unexplained.

In summary, we demonstrate a simple method to decompose QK spaces to study their inner structures.

[1]Harvard University [2]Technion - Israel Institute of Technology [3]Kempner Institute, Harvard University [4]Google DeepMind[*] ([*]Work done entirely at Harvard). Correspondence to: Andrew Lee <ajyl@umich.edu>.

*Proceedings of the 43rd International Conference on Machine Learning*, Seoul, South Korea. PMLR 306, 2026. Copyright 2026 by the author(s).

## 2. Toy Model for QK Decomposition

To motivate our study of QK feature decompositions, we design a simple payload retrieval task. We use italic letters $(a, B)$ for scalars, bold lowercase $(\mathbf{q}, \mathbf{k})$ for vectors, and bold uppercase $(\mathbf{W})$ for matrices. For a brief review of attention heads, see Appendix A.

### 2.1. Task: Payload Retrieval from Context

In our task, an attention head is given a set of payload embeddings, each of which contains some *payload* information (e.g., class label). The model is then given a "selector" embedding, which the model must use to attend to the correct payload embedding and retrieve the correct payload information. Concretely, we generate data of the form $(\mathbf{x}_{1:T}, \mathbf{x}_q, i^*, y_{i^*})$, where $\mathbf{x}_i \in \mathbb{R}^d$ is the payload embedding for timestep $i \in \{1, \ldots, T\}$, $\mathbf{x}_q \in \mathbb{R}^d$ is the selector embedding, $i^*$ is the target timestep to retrieve the payload from, and $y_{i^*} \in \{1, \ldots, P\}$ is the correct payload label that the model must predict. We study two variants of this task, in which embeddings are generated as follows.

**Variant 1: Discrete Latent Variables.** Our data generation relies on $K$ latent variables. For simplicity we set $K = 2$. Our latent variables are each binary sign vectors of length $r_1$ and $r_2$: $\mathbf{z}_1 \in \{-1, 1\}^{r_1}, \mathbf{z}_2 \in \{-1, 1\}^{r_2}$. We refer to them as *latent keys*. Each payload embedding $\mathbf{x}_i$ is generated by first randomly sampling latent keys $\mathbf{z}_{1,i}, \mathbf{z}_{2,i}$ independently, which are then mapped to the embedding space via linear maps $\mathbf{A}_1 \in \mathbb{R}^{d \times r_1}, \mathbf{A}_2 \in \mathbb{R}^{d \times r_2}$, each of which are randomly sampled and fixed from a standard Gaussian. Each payload embedding is also assigned a random payload $y_i \in \{1, \ldots, P\}$, which is also mapped to the embedding space via a fixed linear map $\mathbf{A}_y \in \mathbb{R}^{d \times P}$. Thus the payload embedding is given by:

$$\mathbf{x}_i = \mathbf{A}_1 \mathbf{z}_{1,i} + \mathbf{A}_2 \mathbf{z}_{2,i} + \mathbf{A}_y \mathbf{e}_{y_i} + \boldsymbol{\epsilon}_i \tag{1}$$

where $\mathbf{e}_{y_i}$ is a one-hot encoding of $y_i$ and $\boldsymbol{\epsilon}$ is standard Gaussian noise.

The selector embedding $\mathbf{x}_q$ is generated similarly. We first randomly select a target timestep $i^* \in \{1, \ldots, T\}$. We then use the same latent keys $\mathbf{z}_{1,i^*}, \mathbf{z}_{2,i^*}$ that were used to construct the payload embedding at timestep $i^*$, but now embed them with a different set of embedding matrices $\mathbf{B}_1 \in \mathbb{R}^{d \times r_1}, \mathbf{B}_2 \in \mathbb{R}^{d \times r_2}$, which are also randomly sampled and fixed from a standard Gaussian. The selector embedding is then given by:

$$\mathbf{x}_q = \mathbf{B}_1 \mathbf{z}_{1,i^*} + \mathbf{B}_2 \mathbf{z}_{2,i^*} + \boldsymbol{\epsilon}_q. \tag{2}$$

Unlike the payload embeddings, the selector embedding does not contain any payload information.

To summarize, the payload and selector embeddings share two sets of latent features, $\mathbf{z}_1$ and $\mathbf{z}_2$, but are embedded via different linear maps. Payload embeddings also contain payload information, and the attention head must attend to the correct payload embedding to retrieve the payload.

**Variant 2: Continuous Latent Variables.** The second variant is similar, except that the latent variables are continuous vectors sampled from a standard Gaussian distribution, i.e., $\mathbf{s}_1 \sim \mathcal{N}(\mathbf{0}, \mathbf{I}_{r_1}), \mathbf{s}_2 \sim \mathcal{N}(\mathbf{0}, \mathbf{I}_{r_2})$. Similarly, payload and selector embeddings are generated as follows:

$$\mathbf{x}_i = \mathbf{A}_1 \mathbf{s}_{1,i} + \mathbf{A}_2 \mathbf{s}_{2,i} + \mathbf{A}_y \mathbf{e}_{y_i} + \boldsymbol{\epsilon}_i \tag{3}$$

$$\mathbf{x}_q = \mathbf{B}_1 \mathbf{s}_{1,i^*} + \mathbf{B}_2 \mathbf{s}_{2,i^*} + \boldsymbol{\epsilon}_q \tag{4}$$

### 2.2. Toy Attention Model

We train a single attention head, i.e., weights $\mathbf{W}_Q, \mathbf{W}_K, \mathbf{W}_V \in \mathbb{R}^{d_{\text{head}} \times d}, \mathbf{W}_O \in \mathbb{R}^{P \times d_{\text{head}}}$. Given a data sample $(\mathbf{x}_{1:T}, \mathbf{x}_q, i^*, y_{i*})$, the forward pass and loss are given by:

$$\mathbf{q} = \mathbf{W}_Q \mathbf{x}_q, \quad \mathbf{k}_i = \mathbf{W}_K \mathbf{x}_i, \quad \mathbf{v}_i = \mathbf{W}_V \mathbf{x}_i,$$

$$\alpha_i = \frac{\exp(\mathbf{q}^\top \mathbf{k}_i / \sqrt{d_{\text{head}}})}{\sum_{j=1}^T \exp(\mathbf{q}^\top \mathbf{k}_j / \sqrt{d_{\text{head}}})}, \quad \mathbf{o} = \mathbf{W}_O \sum_{i=1}^T \alpha_i \mathbf{v}_i,$$

$$\hat{y} = \text{softmax}(\mathbf{o}), \quad \mathcal{L} = \text{CrossEntropy}(\hat{y}, y_{i^*}).$$

Thus the model must use $\mathbf{W}_Q, \mathbf{W}_K$ to attend to the correct payload embedding $\mathbf{x}_{i^*}$ and use $\mathbf{W}_V, \mathbf{W}_O$ to decode the correct payload information.

## 3. QK Decomposition using Contrastive Covariance

Here we describe our method of recovering the ranks and subspaces of latent variables in the attention head's query and key spaces. More succinctly, we refer to their bilinear joint embedding space as the *QK space*. One can think of the QK space ($\in \mathbb{R}^{d_{\text{head}} \times d_{\text{head}}}$) as the space of all possible interactions between queries and keys. Note that in all of our analyses, one can replace all instances of $\mathbf{z}_{1,2}$ with $\mathbf{s}_{1,2}$.

Our method constructs a *contrastive covariance* matrix $\Delta\mathbf{C}$ between queries and keys that isolates their interactions attributable to a single latent variable. For instance, consider latent variable $\mathbf{z}_1$. For a sampled query vector $\mathbf{q}$ (associated with target value $\mathbf{z}_{1,i^*}$), we construct two keys:

- $\mathbf{k}_{(\mathbf{z}_1)}^+$, whose $\mathbf{z}_1$ value matches the query ($\mathbf{z}_1 = \mathbf{z}_{1,i^*}$)
- $\mathbf{k}_{(\mathbf{z}_1)}^-$, whose $\mathbf{z}_1$ value differs ($\mathbf{z}_1 \neq \mathbf{z}_{1,i^*}$).

Crucially, we hold $\mathbf{z}_2$ fixed across the two conditions: both keys share the same value of $\tilde{\mathbf{z}}_2$ (drawn randomly) for $\mathbf{z}_2$.

Given a large sample of such triplets $(\mathbf{q}, \mathbf{k}_{(\mathbf{z}_1)}^+, \mathbf{k}_{(\mathbf{z}_1)}^-)$, we

compute *positive* and *negative* covariances:

$$\mathbf{C}^{+}_{(\mathbf{z}_1)} := \mathbb{E}[\mathbf{q}\mathbf{k}^{\top}|+] \in \mathbb{R}^{d_{head} \times d_{head}} \tag{5}$$

$$\mathbf{C}^{-}_{(\mathbf{z}_1)} := \mathbb{E}[\mathbf{q}\mathbf{k}^{\top}|-] \in \mathbb{R}^{d_{head} \times d_{head}} \tag{6}$$

We use the term "covariance" informally, as we are not mean-centering $\mathbf{q}, \mathbf{k}$. Intuitively, $\mathbf{C}^{+}_{(\mathbf{z}_1)}$ captures query-key correlations when $\mathbf{z}_1$ matches, while $\mathbf{C}^{-}_{(\mathbf{z}_1)}$ captures correlations when $\mathbf{z}_1$ does not match. Importantly, because $\mathbf{z}_2$ is held constant across the two conditions, the difference of the covariance terms, $\Delta\mathbf{C}_{(\mathbf{z}_1)}$, isolates the component of query-key interactions that is specifically due to the matching of latent variable $\mathbf{z}_1$ (see Appendix B for the derivation):

$$\Delta\mathbf{C}_{(\mathbf{z}_1)} := \mathbf{C}^{+}_{(\mathbf{z}_1)} - \mathbf{C}^{-}_{(\mathbf{z}_1)}$$
$$= \mathbf{W}_Q\mathbf{B} \begin{bmatrix} \mathbb{E}[\mathbf{z}_{1,i^*}\,\mathbf{z}^{\top}_{1,i^*}] - \mathbb{E}[\mathbf{z}_{1,i^*}\mathbf{z}^{\top}_{1,i\neq i^*}] & \mathbf{0} \\ \mathbf{0} & \mathbf{0} \end{bmatrix} \mathbf{A}^{\top}\mathbf{W}^{\top}_K$$

where $\mathbf{B} := [\mathbf{B}_1, \mathbf{B}_2]$ and $\mathbf{A} := [\mathbf{A}_1, \mathbf{A}_2]$. The same procedure can be repeated for $\Delta\mathbf{C}_{(\mathbf{z}_2)}$ by defining positive and negative conditions accordingly.

**Recovering the ranks and subspaces of latent variables.** Given $\Delta\mathbf{C}_{(\mathbf{z}_1)}$, we can recover the rank and subspace of latent variable $\mathbf{z}_1$ by performing SVD:

$$\Delta\mathbf{C}_{(\mathbf{z}_1)} = \mathbf{U}_{(\mathbf{z}_1)}\mathbf{\Sigma}_{(\mathbf{z}_1)}\mathbf{V}^{\top}_{(\mathbf{z}_1)} \tag{7}$$

The rank of $\mathbf{z}_1$ (denoted $r_1$) can be estimated by counting the number of singular values that capture 99% of the squared Frobenius norm of $\Delta\mathbf{C}_{(\mathbf{z}_1)}$. Denoting the top-$r_1$ singular vectors as $\mathbf{U}^{[:r_1]}_{(\mathbf{z}_1)}$ and $\mathbf{V}^{[:r_1]}_{(\mathbf{z}_1)}$, $\mathbf{U}^{[:r_1]}_{(\mathbf{z}_1)}$ gives a basis in query space that encodes $\mathbf{z}_1$, while $\mathbf{V}^{[:r_1]}_{(\mathbf{z}_1)}$ gives a basis in key space. This can be repeated for each latent variable to recover their respective ranks and subspaces.

## 4. Empirical Validation of QK Decomposition

Here we apply our method on attention heads trained on the payload retrieval task.

**Experimental Setup.** We train a single attention head under various task settings and hyperparameters. We study both task variants (Section 2.1): discrete $(\mathbf{z}_1, \mathbf{z}_2)$ and continuous $(\mathbf{s}_1, \mathbf{s}_2)$ latent variables. We train attention heads with either $d_{\text{head}} = 8$, while varying $r_1, r_2 \in \{2, \dots, 6\}$, or with $d_{\text{head}} = 16$ with $r_1, r_2 \in \{4, \dots, 12\}$. In every settings, we set $d = 32$, context length $T = 16$, and the number of payloads (classes) $P = 10$. Under these settings, the attention heads achieve 99% accuracy, except for the continuous task when $d_{\text{head}} = 8$, in which accuracy drops to around 85%. For additional training details, see Appendix D.

**Recovering Rank of Latent Variables.** We first verify that our method recovers the rank of each latent variable.

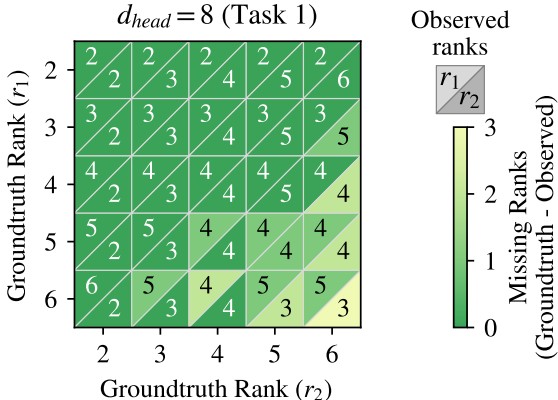

*Figure 2.* **Contrastive QK decomposition recovers the groundtruth rank of each latent variable**, as long as there is no superposition (i.e., $r_1 + r_2 < d_{\text{head}}$). Each cell annotates the recovered ranks $r_1, r_2$, while the x and y-axes indicate the groundtruth ranks. The color of each cell indicates the difference between groundtruth and recovered ranks.

Figure 2 shows the results for one of our models (all other results in Appendix F). The x, y-axes indicate the groundtruth ranks of $\mathbf{z}_1$ or $\mathbf{z}_2$, $r_1$ and $r_2$. The text annotations indicate the ranks recovered by our method. The colors indicate the difference between the groundtruth and recovered ranks.

When the model has enough dimensions to encode both latent variables ($r_1 + r_2 < d_{\text{head}}$), our method recovers the ranks of both latent variables (dark green cells). Otherwise, we see *superposition* (Elhage et al., 2022), in which the model compresses both variables using less dimensions than available. We discuss superposition in more detail below.

**Recovering Latent Variable Subspaces in QK Space.** We can apply SVD on $\Delta\mathbf{C}$ to recover the subspaces in which each latent variable is encoded. As a reminder, we denote the top-$r_1$ singular vectors of $\Delta\mathbf{C}_{(\mathbf{z}_1)}$ as $\mathbf{U}^{[:r_1]}_{(\mathbf{z}_i)} \in \mathbb{R}^{d_{\text{head}} \times r_1}$ and $\mathbf{V}^{[:r_1]}_{(\mathbf{z}_1)} \in \mathbb{R}^{d_{\text{head}} \times r_1}$. $\mathbf{U}^{[:r_1]}_{(\mathbf{z}_1)}$ provides a basis for $\mathbf{z}_1$ in query space, while $\mathbf{V}^{[:r_1]}_{(\mathbf{z}_1)}$ provides a basis in key space.

We visualize these subspaces by projecting the query, key vectors $\mathbf{q}, \mathbf{k} \in \mathbb{R}^{d_{\text{head}}}$ onto $\mathbf{U}^{[:r_1]}_{(\mathbf{z}_1)}$ and $\mathbf{V}^{[:r_1]}_{(\mathbf{z}_1)}$, followed by PCA. Figure 3 shows an example for a model with $d_{\text{head}} = 16$ and $r_1 = 3, r_2 = 5$. Note two observations: first, $\mathbf{z}_1$ is sampled from $\{-1, 1\}^{r_1}$ which corresponds to the vertices of a 3D cube, which is faithfully recovered from PCA. Second, the key and query projections are aligned, both of which collapse to the same clusters. For an example of the second task variant, see Figure 12, in which we recover the Gaussian sphere structure of latent key $\mathbf{s}_1$.

**Causal Interventions in QK Space.** To validate the role of the recovered subspaces, we perform causal interventions. Namely, we intervene on the key vectors by first projecting them onto their latent variable subspaces. We then change

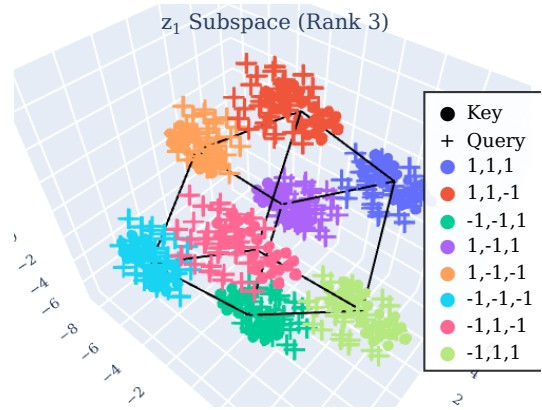

z₁ Subspace (Rank 3)

*Figure 3.* **PCA of Latent Variable Subspace.** We project key and query vectors onto the recovered subspaces of latent variable $\mathbf{z}_1$ (of rank $r_1 = 3$), then perform PCA, which recovers the 3D-cube structure of $\mathbf{z}_1$. Also note that keys and queries align onto the same clusters. See Figure 12 for the continuous task variant, in which our method recovers the spherical structure of latent variable $\mathbf{s}_1$.

the coordinates in these subspaces (imagine moving from one vertex to another in Figure 3), and measure how the attention scores change.

More specifically, consider intervening on $\mathbf{z}_1$. Given an original timestep $i_{\text{orig.}}$, we randomly select a new target timestep $i_{\text{target}}$. We then project the key vectors $\mathbf{k}_{i_{\text{orig.}}}, \mathbf{k}_{i_{\text{target}}}$ onto the subspaces of $\mathbf{z}_1$, then replace the coordinates of $\mathbf{k}_{i_{\text{orig.}}}$ in these subspaces with those of $\mathbf{k}_{i_{\text{target}}}$, and vice versa:

$$\mathbf{P_v} = \mathbf{V}_{(\mathbf{z}_1)}^{[:r_1]}\mathbf{V}_{(\mathbf{z}_1)}^{[:r_1]\top}, \tag{8}$$

$$\tilde{\mathbf{k}}_{i_{\text{orig.}}} = \mathbf{k}_{i_{\text{orig.}}} + \mathbf{P_v}\,(\,\mathbf{k}_{i_{\text{target}}} - \mathbf{k}_{i_{\text{orig.}}}\,) \tag{9}$$

$$\tilde{\mathbf{k}}_{i_{\text{target}}} = \mathbf{k}_{i_{\text{target}}} + \mathbf{P_v}(\,\mathbf{k}_{i_{\text{orig.}}} - \mathbf{k}_{i_{\text{target}}}\,) \tag{10}$$

Finally, we compute attention scores with these modified keys and measure how much of the attention score has shifted from timestep $i_{\text{orig.}}$ to $i_{\text{target}}$. Note that this step can be repeated using $\mathbf{z}_2$ to intervene on both latent variables.

Figure 4 shows the results on a test set of 51,200 samples (for more examples see Appendix F). $z_1$, $z_2$, and $z_1 + z_2$ correspond to intervening on $\mathbf{V}_{(\mathbf{z}_1)}^{[:r_1]}, \mathbf{V}_{(\mathbf{z}_2)}^{[:r_2]}$, or both. "Rand $r_1$, $r_2$, $r_1+r_2$" correspond to intervening on random subspaces of the same dimension as $\mathbf{z}_1$ or $\mathbf{z}_2$. Note that intervening on both subspaces ($z_1 + z_2$) moves all the attention from $i_{\text{orig.}}$ to $i_{\text{target}}$, while intervening on the random baseline counterparts induces a much smaller shift. This validates that our QK decomposition method recovers the correct subspaces in which latent variables are encoded.

**Pitfalls of Contrastive Covariance: Feature Splits and Superposition.** Our toy model also reveals pitfalls of our QK decomposition method. To illustrate them, we study how our latent variables interact with each other in QK space by analyzing their bilinear interactions $\mathbf{G}$:

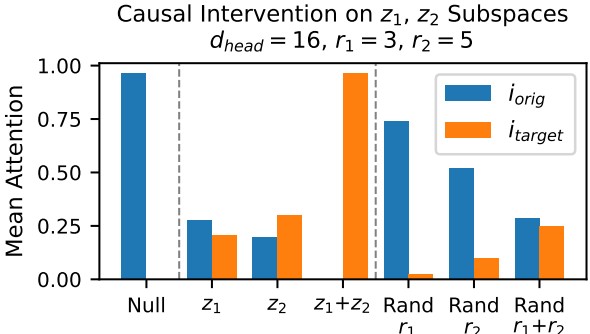

Causal Intervention on $z_1$, $z_2$ Subspaces
$d_{head} = 16$, $r_1 = 3$, $r_2 = 5$

*Figure 4.* **Causal Interventions on Latent Variable Subspaces.** Intervening on the recovered subspaces for latent variables $\mathbf{z}_1$ and $\mathbf{z}_2$ shifts all the attention from the original token to the target token, while intervening on random subspaces of the same dimension (i.e., "Rand $r_1, r_2, r_1 + r_2$") has less of an effect.

$$\mathbf{q}^\top\mathbf{k} = (\mathbf{W}_Q\mathbf{B}\mathbf{z}_q)^\top(\mathbf{W}_K\mathbf{A}\mathbf{z}_k) \tag{11}$$

$$= \mathbf{z}_q^\top \underbrace{\mathbf{B}^\top\mathbf{W}_Q^\top\mathbf{W}_K\mathbf{A}}_{\mathbf{G}}\,\mathbf{z}_k\ = \mathbf{z}_q^\top\mathbf{G}\mathbf{z}_k \tag{12}$$

where $\mathbf{A} := [\mathbf{A}_1, \mathbf{A}_2], \mathbf{B} := [\mathbf{B}_1, \mathbf{B}_2] \in \mathbb{R}^{d \times (r_1+r_2)}$, $\mathbf{z}_q = [\mathbf{z}_{1,i^*}; \mathbf{z}_{2,i^*}], \mathbf{z}_k = [\mathbf{z}_1; \mathbf{z}_2] \in \mathbb{R}^{r_1+r_2}$. Here, $\mathbf{G} \in \mathbb{R}^{(r_1+r_2)\times(r_1+r_2)}$ captures the bilinear interactions between each latent variable in the query and key spaces, where $\mathbf{G}[i,j]$ indicates how strongly latent variables $\mathbf{z}_{q,i}$ and $\mathbf{z}_{k,j}$ interact, via weights $\mathbf{W}_Q^\top\mathbf{W}_K$.

Figure 5 visualizes $\mathbf{G}$ under varying $d_{\text{head}}$ sizes and ranks of each latent variable, for the second task variant. For results on the first task, see Appendix F.

We make two observations. First, when the model has enough dimensions to represent both latent variables ($r_1 + r_2 \leq d_{\text{head}}$), we observe *feature splits*, as indicated by the strong diagonals in such settings. Namely, while our latent variables $\mathbf{z}_1, \mathbf{z}_2$ have $r_1, r_2$ degrees of freedom, their coordinates (e.g., $\mathbf{z}_1[0], \mathbf{z}_1[1]$) are independent of one another. Thus the model further decomposes these latent variables into independent components.

On the contrary, with not enough dimensions ($r_1 + r_2 > d_{\text{head}}$), we observe *superposition*, where the model compresses both latent variables using fewer dimensions. This is indicated by the off-diagonal interactions in $\mathbf{G}$, where multiple components from $\mathbf{z}_1$ and $\mathbf{z}_2$ interact. The subsequent softmax operation likely allows such compression to occur with its "winner-takes-all" behavior. This raises two questions: how often does superposition occur in "real" models, and how do we interpret superposed features?

**So What is a Feature?** Note that our method relies on a *human-defined* notion of what constitutes as a "feature", which is manifested in how the positive and negative covariance conditions are defined. Though our method faithfully recovers the targeted latent variables as designed by our positive and negative pairs, this human-defined notion of

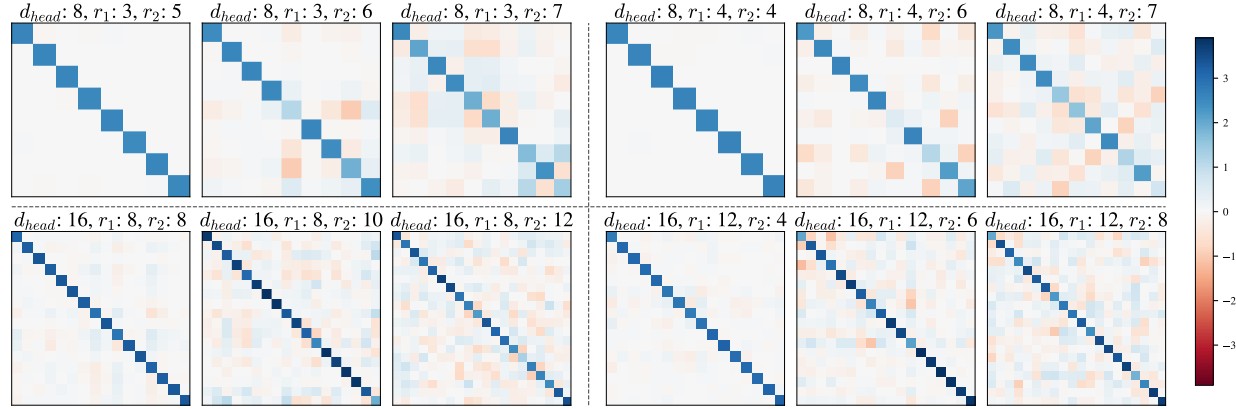

*Figure 5.* **Interactions between latent variables in QK space reveal feature splits and superposition.** When the model has enough dimensions ($r_1 + r_2 \leq d_{\text{head}}$), the model further decomposes the latent variables into independent components (feature splits: strong diagonals in **G**, as opposed to block diagonals). When there are not enough dimensions ($r_1 + r_2 > d_{\text{head}}$), we observe *superposition*, in which the model compresses both latent variables into fewer dimensions than available (off-diagonal interactions in **G**).

features may not always align with the "unit" in which the model represents features, as our examples demonstrate. All of this adds to the on-going discourse around "what is a feature?" (Olah et al., 2020; Elhage et al., 2022).

## 5. QK Features in Large Language Models

Here we apply our method to Llama 3.1-8B Instruct (Grattafiori et al., 2024) and Qwen 3-4B Instruct (Yang et al., 2025). Results for Qwen are in Appx G.

### 5.1. Categorical Semantic Space in Filter Heads

Filter Heads (Sharma et al., 2025) refer to attention heads that mirror "filter" functions: for instance, given a list of items, they attend to items pertaining to a queried category:

```
Cat, apple, dog, truck, orange,
tea, car, duck.  Find the fruits.
```

We apply our method to identify QK subspaces that encode various categories in Filter Heads.

To do so, we emulate the setup of Sharma et al. to identify Filter Heads. We construct 2,000 prompts containing a list of items from various categories $c \in \mathcal{C}$ (e.g., fruits, animals, vehicles), followed by a query category $c^*$. Each prompt includes at least 5 items per category. We select the top three heads based on the ratio of attention given to the queried items versus all other items.

We use the last token for our query vector, and use key vectors for positive and negative QK covariances as defined below (per category):

- $\mathbf{C}^+_{\text{category}}$: tokens belonging to the queried category $c^*$.
- $\mathbf{C}^-_{\text{category}}$: tokens *not* belonging to the queried category.

The remaining steps follow as in Section 3.

**Visualizing Categorical Semantic QK Space.** We provide two visualizations of the recovered categorical semantic space. In the first, we consider 5 categories: fruits, animals, vehicles, drinks, and countries. Interestingly, their contrastive covariances ($\Delta\mathbf{C}_{\text{fruits}}, \Delta\mathbf{C}_{\text{animals}}, \dots$) all result in rank 1. We thus define the categorical QK subspace as the span of these 5 directions.

Figure 6a visualizes the keys and queries projected onto this categorical subspace using PCA. We observe clear clusters corresponding to each category, but more importantly, we also observe alignment between keys and queries of the same category. Namely, the first principal component (PC 1) separates keys from queries, while the structure of queries and keys in PC 2 and 3 are symmetric to one another. In Figure 6b we expand the list of categories to 13 and visualize only the keys, which again reveals clear semantic clusters.

**Causal Interventions.** We validate the role of the identified subspace with interventions. We use a test set of 1,000 samples, each of which has 5 categories. In each sample, we randomly select a target token $i_{target}$ that does *not* belong to the queried category. We then intervene on the recovered subspaces as described in Equations (9), (10). Figure 7 shows that intervening on the recovered 5-dimensional subspace successfully shifts attention from one categorical token to another (e.g., from fruits to animals), and is much more effective than a random 5-dimensional baseline. It does not, however, shift all the attention, suggesting additional features in QK space not captured by our method.

### 5.2. Binding Features

Researchers have studied how language models bind entities together (Feng & Steinhardt, 2023; Dai et al., 2024; Prakash et al., 2025). Gur-Arieh et al. (2025) show that models rely on multiple mechanisms. Consider the following prompt:

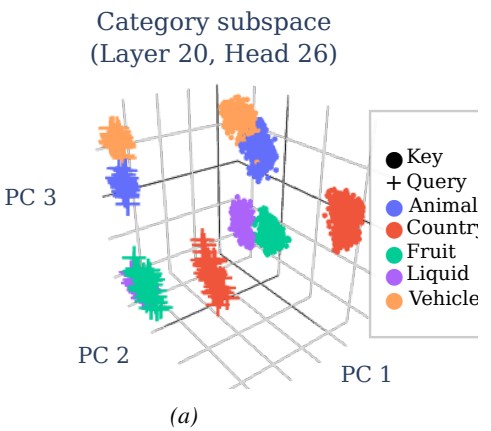
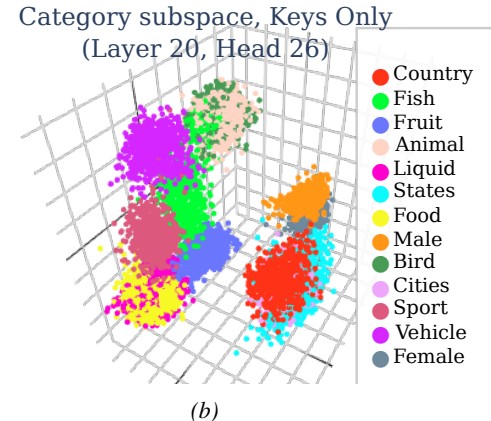

*(a)*          *(b)*

*Figure 6.* **(a) PCA visualization of the categorical QK subspace.** We project key and query vectors onto their respective categorical subspaces and perform PCA. Note the alignment between keys and queries of the same category. **(b) PCA visualization of additional categories (keys only).** Visualizing additional categories exhibits clear semantic clusters (e.g., locations (Country, States, Cities), names (Male, Female), animals (Animal, Bird), food (Food, Liquid, Fruit)).

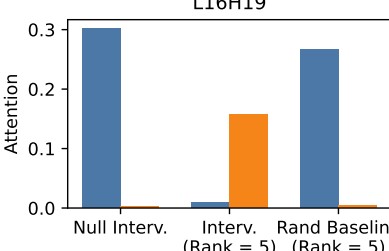
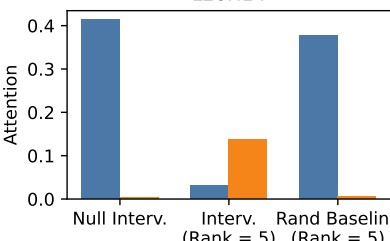
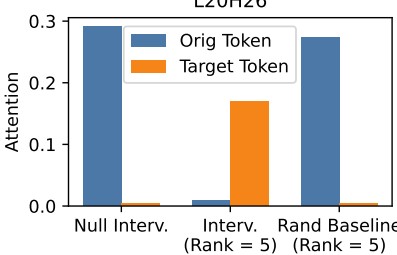

*Figure 7.* **Causal interventions on categorical QK subspaces.** We intervene by replacing the QK components of tokens from one category (e.g., fruits) with those from another category (e.g., animals).

```
The hat is in box O. The jam is in box Z
...(omitted)...Which box is the jam in?
```

One mechanism is dubbed *order-ID*, in which the model uses the order in which entity groups appear: given a query entity (e.g., jam), the model retrieves the box with the same order (e.g., second) as the queried entity. Another mechanism is the *lexical* mechanism: the model uses the identity of the queried entity (e.g., jam) to retrieve the associated box. This is perhaps the most intuitive, "correct" mechanism. For more details on these mechanisms, see Appendix E.

We use our method to identify QK subspaces corresponding to these two mechanisms. We construct 3,000 prompts, each containing 9 entity-box pairs (e.g., hat-box O, jam-box Z, etc.). We filter for attention heads that attend to the correct box with at least 30% accuracy. This results in 9 heads - we demonstrate results from a few heads here while all others can be found in Appendix F. We use the last token as our query and box label tokens (e.g., box "Z") as our keys.

For order-ID, the positive and negative covariances are:

- $\mathbf{C}_{\text{order}}^{+}$: box whose order matches that of the queried entity.
- $\mathbf{C}_{\text{order}}^{-}$: boxes whose order does not match that of the queried entity.

Importantly, we keep the same set of entities in all of our samples (although their orders are shuffled across samples), and use the same *fixed* query entity across all samples. However, in our intervention test data, we use query entities *not seen* when constructing $\Delta\mathbf{C}_{\text{order}}$.

For the lexical mechanism, we make counterfactual prompts: for every prompt, we make a copy but replace the entity being queried ("...the jam is in box Z...Which box is the jam in?" → "...the pen is in box Z...Which box is the pen in?"). Our positive and negative covariances are defined as:

- $\mathbf{C}_{\text{Lex.}}^{+}$: box of the original queried entity.
- $\mathbf{C}_{\text{Lex.}}^{-}$: box of the queried entity in counterfactual prompt.

Similar to the order-IDs, this allows us to isolate signals coming from lexical information.

**Visualizing Binding QK Subspaces.** Here we visualize our recovered binding QK subspaces. We use 3,000 samples using 9 entities each to construct $\Delta\mathbf{C}_{\text{order}}$ and $\Delta\mathbf{C}_{\text{Lex.}}$. We find that $\Delta\mathbf{C}_{\text{order}}$ is usually rank 2 or 3, while $\Delta\mathbf{C}_{\text{Lex.}}$ is usually rank 9 or 10 (the ranks do not appear to depend on the number of entities used in constructing $\Delta\mathbf{C}$ – see Figure 16). We project our keys and query vectors onto these respective subspaces and visualize them using PCA or

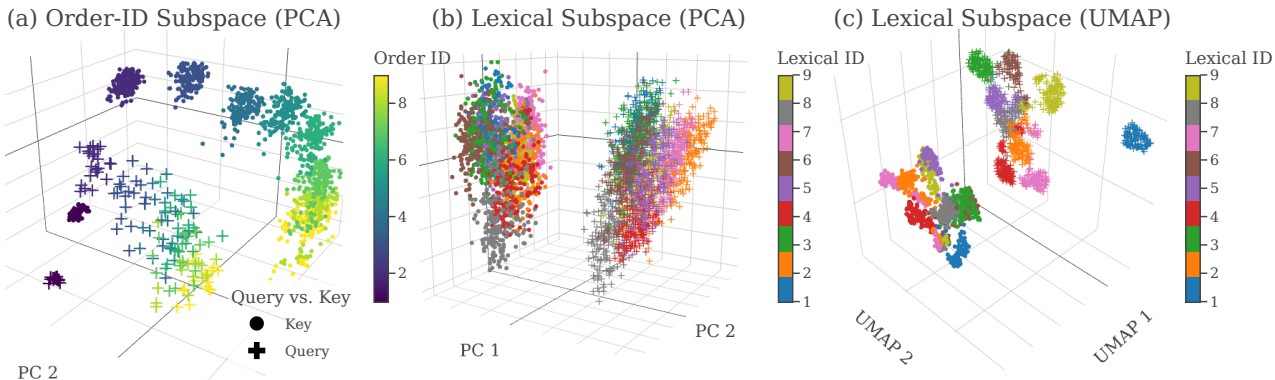

Figure 8. **PCA, UMAP of order-ID and lexical subspaces.** PC1/UMAP1 encode keys versus queries, while PCs/UMAPs 2 and 3 encode order or lexical IDs. Note the alignment between keys and queries in order-IDs. Because the lexical subspace is higher dimensional, we include both PCA and UMAP: the clusters are easier to see in UMAP, while the alignment between keys and queries is easier to see in the PCA (note that UMAP does not preserve the notion of distance, and thus alignment information is not visually observable). Visualizing the same PCAs on key and query vectors *without* projecting to our QK subspaces reveals that order-ID features are encoded in the first few PCs (see Figure 17).

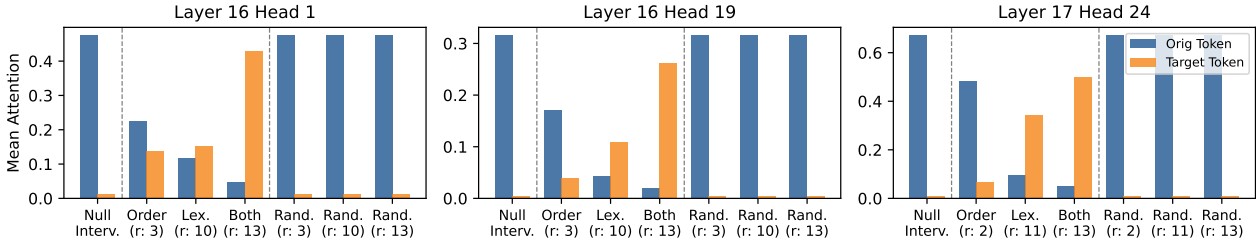

Figure 9. **Causal interventions on binding QK subspaces.** We intervene by modifying the order-ID or lexical components (or both) of the QK space. Intervening on both components yields a larger shift in attention.

UMAP. Because the lexical subspace has more dimensions, we include a UMAP visualization. Figure 8 shows the results. Similar to categorical features, we observe clear clusters corresponding to order-IDs and lexical-IDs, as well as alignment between keys and queries.

**Causal Interventions.** We further do causal interventions on these binding QK subspaces. We use 1,000 test samples. Similar to previous experiments, given an original timestep $i_{orig}$ corresponding to the correct box, we select a random target timestep $i_{target}$ corresponding to a different box. We then intervene the key vectors of $\mathbf{k}_{i_{orig}}$ and $\mathbf{k}_{i_{target}}$ in either the order-ID subspace, lexical subspace, or both. Results are shown in Figure 9, in which we see a similar trend as before: intervening on each individual subspace can shift some of the attention, while intervening on both subspaces shifts the majority of the attention. Intervening on random subspaces of the same ranks has negligible effects.

### 5.3. Attention Logit Attributions

How much of the attention logits (attention scores prior to softmax) can be explained by our recovered features, and how much is left unexplained? Because the logits are linear in query space, we can easily check how much our features contribute towards an attention head's logits.

Namely, given $\mathbf{q}, \mathbf{k}_i \in \mathbb{R}^{d_{\text{head}}}$ for key positions $i \in \{1, \dots, T\}$, let $\mathbf{K} \in \mathbb{R}^{T \times d_{\text{head}}}$ be the stacked matrix of keys, with each row $\mathbf{K}[i] = \mathbf{k}_i^\top$. The pre-softmax attention logits are $\ell = \mathbf{K}\mathbf{q}/\sqrt{d_{\text{head}}} \in \mathbb{R}^T$.

Now consider our recovered feature basis for order-ID and lexical-ID in query space: $\mathbf{U}_{(\text{order})}$, $\mathbf{U}_{(\text{Lex.})}$, each of rank $r_{\text{order}}, r_{\text{Lex.}} \ll d_{\text{head}}$. Let $\mathbf{P}_{\text{order}} := \mathbf{U}_{\text{order}}\mathbf{U}_{\text{order}}^\top$ be an orthogonal projector. Intuitively, $\mathbf{P}_{\text{order}}\mathbf{q} \in \mathbb{R}^{d_{\text{head}}}$ is the subspace in $\mathbf{q}$ that encodes order-ID, as everything else orthogonal to the column space of $\mathbf{U}_{\text{order}}$ is removed. Define a similar orthogonal projection $\mathbf{P}_{\text{Lex.}}$ for lexical ID, and we can iteratively decompose our query vector:

$$\mathbf{q}_{\text{order}} = \mathbf{P}_{\text{order}}\mathbf{q}, \tag{13}$$

$$\mathbf{q}_{\text{Lex.}} = \mathbf{P}_{\text{Lex.}}\big(\mathbf{q} - \mathbf{q}_{\text{order}}\big), \tag{14}$$

$$\mathbf{q}_\perp = \mathbf{q} - \mathbf{q}_{\text{order}} - \mathbf{q}_{\text{Lex.}}, \tag{15}$$

where $\mathbf{q}_{\text{Lex.}}$ identifies the lexical subspace in $\mathbf{q}$ *after* the order-ID subspace has been removed, and $\mathbf{q}_\perp$ is the residual query space that is not accounted for by order-ID and lexical-ID. By construction, $\mathbf{q} = \mathbf{q}_{\text{order}} + \mathbf{q}_{\text{Lex.}} + \mathbf{q}_\perp$. Note that when the two feature subspaces are not distinct, this decomposition is sensitive to the order in which we project out feature subspaces, as the overlapping space will count

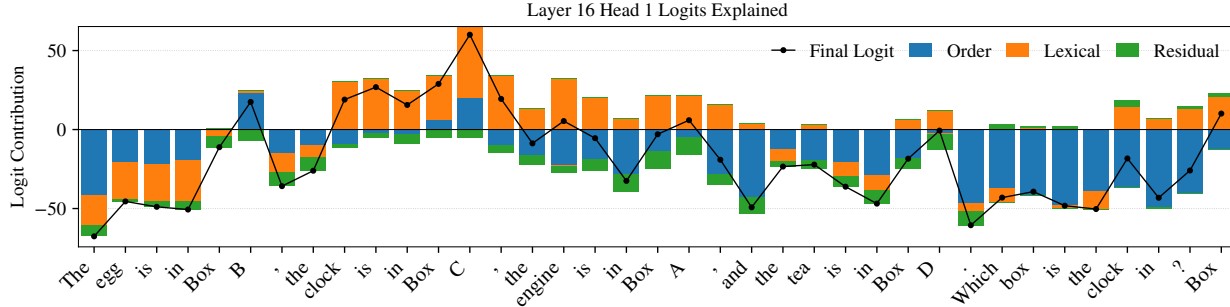

*Figure 10.* **Attention logit attributions to low-rank feature components.** Blue and orange bars refer to logit contributions from the order-ID and lexical subspaces. The green bars indicate logits left unexplained by our two features.

towards the first feature. In our case we project out $\mathbf{U}_{\text{order}}$ first because it has fewer ranks than $\mathbf{U}_{\text{Lex.}}$.

Finally, with our decomposed query vectors, we can also define feature-specific logit vectors:

$$\ell_{\text{order}} = \frac{\mathbf{K}\mathbf{q}_{\text{order}}}{\sqrt{d_{\text{head}}}}, \quad \ell_{\text{Lex.}} = \frac{\mathbf{K}\mathbf{q}_{\text{Lex.}}}{\sqrt{d_{\text{head}}}}, \quad \ell_{\perp} = \frac{\mathbf{K}\mathbf{q}_{\perp}}{\sqrt{d_{\text{head}}}}.$$

Because the logit space is linear in $\mathbf{q}$, we have the following decomposition:

$$\ell = \ell_{\text{order}} + \ell_{\text{Lex.}} + \ell_{\perp}, \quad \ell_i = \ell_i^{(\text{order})} + \ell_i^{(\text{Lex.})} + \ell_i^{(\perp)} \quad \forall i.$$

where $\ell_i$ is the logit at timestep $i$. This yields token-level attributions in logit space: $\ell_i^{(\text{order})}$ and $\ell_i^{(\text{Lex.})}$ measure how much that token's logit is accounted for by the recovered order-ID vs. lexical subspaces, with $\ell_i^{(\perp)}$ capturing the residual contribution not explained by these subspaces.

Figure 10 demonstrates an example: given an input sentence, per token, blue and orange bars indicate logits attributable to the order-ID and lexical subspaces, while green bars indicate residual logits that are left unexplained. In addition to attention logits left unexplained, this example provides a couple more insights. For instance, this head seems to rely on lexical-IDs more than order-IDs, although this may be a result of the lexical subspace having higher rank. We can also observe mistakes that may have gone unnoticed (especially post-softmax), as we see the model incorrectly assigning mass onto the order-ID subspace of Box B, or the lexical subspace of Box A.

## 6. Related Work

Here we provide an abridged overview of prior work, with a much more thorough review in Appendix C.

QK spaces have been studied before, in both language and vision models. In language, Kamath et al. (2025), Ge et al. (2024), and Friedman et al. (2025) decompose query-key interactions using features from sparse autoencoders, while Gurnee et al. (2026) use features from probes to study their interactions in QK space. Lastly, Wynrow & Sharkey (2024)

learn a sparse mask in QK space to detect features. Unlike prior work, our method does not rely on pre-existing features, nor any training, in order to find QK features.

In vision, Pan et al. (2024) and Doshi et al. (2026) similarly apply SVD on query-key interactions, finding "channels" that communicate positional or content information, while Li et al. (2025) study how vision models bind tokens belonging to the same entity via bilinear probes in QK space.

Researchers have also viewed attention as a "communication channel" (Elhage et al., 2021). Merullo et al. (2024) studies heads that "talk" with one another, while Franco & Crovella (2025) recovers low-rank QK subspaces that are causally relevant for upstream usage within a circuit.

Lastly, researchers have also studied attention heads by visualizing query-key interactions (Yeh et al., 2023), uncovering global patterns in their interactions.

## 7. Discussion

We demonstrate a simple method to decompose the QK space of attention heads into interpretable low-rank components. Here we briefly discuss potential future directions.

**Multi-dimensional Features.** In our work and others (Engels et al., 2025), we have seen multi-dimensional features. How might we detect other multi-dimensional features?

**Unsupervised QK Decomposition.** One limitation of our method is its reliance on positive and negative covariance terms, which requires knowing what features to look for beforehand. A natural next step may be decomposing QK spaces without human supervision. One potential challenge may be in dealing with multi-dimensional features of *varying ranks*. Another challenge may be in interpreting such decomposed components: even if we identify multiple QK components, their observable behaviors may be identical (e.g., they both attend to token X). When multiple components exhibit the same behavior, how might we interpret each component? We leave these questions to future work.

Code: `https://github.com/ajyl/QK`

## Impact Statement

This paper takes a step towards interpreting the internal computations of large language models. We hope such interpretable systems will lead to safer and more reliable use cases in the future.

## Acknowledgements

AL thanks Eric Todd, Andy Arditi, Sheridan Feucht, and Yida Chen for constructive feedback. AL acknowledges support from a Superalignment Fast Grant from OpenAI. YB was funded by Coefficient Giving, the Israel Science Foundation (grant No. 2942/25), and the European Union (ERC, Control-LM, 101165402). Views and opinions expressed are however those of the authors only and do not necessarily reflect those of the European Union or the European Research Council Executive Agency. Neither the European Union nor the granting authority can be held responsible for them. Lastly, FV and MW acknowledge support from a Superalignment Fast Grant from OpenAI, and Coefficient Giving.

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

## A. Attention Review

We use lowercase letters $(a, b)$ for scalars, bold lowercase $(\mathbf{q}, \mathbf{k})$ for vectors, bold uppercase $(\mathbf{W})$ for matrices.

Consider a single attention head with key, query, and value weight matrices

$$\mathbf{W}_K, \ \mathbf{W}_Q, \ \mathbf{W}_V \in \mathbb{R}^{d_{\text{head}} \times d}.$$

At position $t$ in the sequence, the activations are $\mathbf{x}_t \in \mathbb{R}^d$ and the head computes

$$\mathbf{q}_t = \mathbf{W}_Q \mathbf{x}_t \in \mathbb{R}^{d_{\text{head}}}, \quad \mathbf{k}_s = \mathbf{W}_K \mathbf{x}_s \in \mathbb{R}^{d_{\text{head}}},$$

$$\ell_{t,s} = \mathbf{q}_t^\top \mathbf{k}_s, \quad \alpha_{t,s} = \frac{\exp\left(\ell_{t,s}/\sqrt{d_{\text{head}}}\right)}{\sum_{s'} \exp\left(\ell_{t,s'}/\sqrt{d_{\text{head}}}\right)},$$

where $\ell_{t,s}$ is the unnormalized attention logit from query position $t$ to key position $s$, and $\alpha_{t,s}$ is the corresponding attention weight.

The logit is a bilinear form in the residual stream:

$$\ell_{t,s} = \mathbf{q}_t^\top \mathbf{k}_s = \mathbf{x}_t^\top \mathbf{W}_Q^\top \mathbf{W}_K \mathbf{x}_s = \mathbf{x}_t^\top \mathbf{B} \mathbf{x}_s,$$

$$\text{rank}(\mathbf{B}) \leq d_{\text{head}} \ll d$$

We are interested in decomposing $\mathbf{x}^\top \mathbf{B} \mathbf{x}$ further into interpretable subspaces that encode specific features.

## B. Contrastive Covariance Derivation

Our contrastive covariance matrix $\Delta \mathbf{C}_{(\mathbf{z}_1)}$ captures the interaction between query and key vectors that is specifically due to the matching of latent variable $\mathbf{z}_1$. To see this, we start with the definitions of key and query terms.

Remember from Equations 1 and 2 that the payload embeddings and selector embeddings are generated as follows:

$$\mathbf{x}_i = \mathbf{A}_1 \mathbf{z}_{1,i} + \mathbf{A}_2 \mathbf{z}_{2,i} + \mathbf{A}_y \mathbf{e}_{y_i} + \boldsymbol{\epsilon}_i$$
$$\mathbf{x}_q = \mathbf{B}_1 \mathbf{z}_{1,i^*} + \mathbf{B}_2 \mathbf{z}_{2,i^*} + \boldsymbol{\epsilon}_q.$$

Thus query and key vectors are given by:

$$\begin{aligned}
\mathbf{q} &= \mathbf{W}_Q \mathbf{x}_q \\
&= \mathbf{W}_Q \left(\mathbf{B}_1 \mathbf{z}_{1,i^*} + \mathbf{B}_2 \mathbf{z}_{2,i^*} + \boldsymbol{\epsilon}_q\right) \\
\mathbf{k}_i &= \mathbf{W}_K \mathbf{x}_i \\
&= \mathbf{W}_K \left(\mathbf{A}_1 \mathbf{z}_{1,i} + \mathbf{A}_2 \mathbf{z}_{2,i} + \mathbf{A}_y \mathbf{e}_{y_i} + \boldsymbol{\epsilon}_i\right)
\end{aligned}$$

Assuming that the attention head's key vectors do not encode payload information (i.e., $\mathbf{W}_K \mathbf{A}_y \approx \mathbf{0}$) and ignoring noise terms, we can express the above in vectoral form:

$$\mathbf{q} = \mathbf{W}_Q \mathbf{B} \begin{bmatrix} \mathbf{z}_{1,i^*} \\ \mathbf{z}_{2,i^*} \end{bmatrix},$$

$$\mathbf{k}_i = \mathbf{W}_K \mathbf{A} \begin{bmatrix} \mathbf{z}_{1,i} \\ \mathbf{z}_{2,i} \end{bmatrix}$$

where $\mathbf{B} := [\mathbf{B}_1 \; \mathbf{B}_2]$ and $\mathbf{A} := [\mathbf{A}_1 \; \mathbf{A}_2]$.

Now consider the positive covariance term $\mathbf{C}^+_{(\mathbf{z}_1)}$. The positive covariance is defined as pairs of $\mathbf{q}, \mathbf{k}$ where the latent variable $\mathbf{z}_1$ matches, while $\mathbf{z}_2$ is held constant (i.e., $\mathbf{z}_{2,i} = \tilde{\mathbf{z}}_2$).

Thus we have:

$$\mathbf{k}_i^+ = \mathbf{W}_K \mathbf{A} \begin{bmatrix} \mathbf{z}_{1,i^*} \\ \tilde{\mathbf{z}}_2 \end{bmatrix},$$

$$\mathbb{E}[\mathbf{q}\mathbf{k}_i^{+\top}|+]$$
$$= \mathbb{E}\left[ \left(\mathbf{W}_Q \mathbf{B} \begin{bmatrix} \mathbf{z}_{1,i^*} \\ \mathbf{z}_{2,i^*} \end{bmatrix}\right) \left(\mathbf{W}_K \mathbf{A} \begin{bmatrix} \mathbf{z}_{1,i^*} \\ \tilde{\mathbf{z}}_2 \end{bmatrix}\right)^\top \right]$$
$$= \mathbb{E}\left[ \mathbf{W}_Q \mathbf{B} \begin{bmatrix} \mathbf{z}_{1,i^*} \\ \mathbf{z}_{2,i^*} \end{bmatrix} [\mathbf{z}_{1,i^*}^\top \tilde{\mathbf{z}}_2^\top] \mathbf{A}^\top \mathbf{W}_K^\top \right]$$

$$= \mathbf{W}_Q \mathbf{B} \mathbb{E}\left[ \begin{bmatrix} \mathbf{z}_{1,i^*} \\ \mathbf{z}_{2,i^*} \end{bmatrix} [\mathbf{z}_{1,i^*}^\top, \tilde{\mathbf{z}}_2^\top] \right] \mathbf{A}^\top \mathbf{W}_K^\top$$

$$= \mathbf{W}_Q \mathbf{B} \begin{bmatrix} \mathbb{E}[\mathbf{z}_{1,i^*}\mathbf{z}_{1,i^*}^\top] & \mathbb{E}[\mathbf{z}_{1,i^*}]\mathbb{E}[\tilde{\mathbf{z}}_{2,i}^\top] \\ \mathbb{E}[\mathbf{z}_{2,i^*}]\mathbb{E}[\mathbf{z}_{1,i^*}^\top] & \mathbb{E}[\mathbf{z}_{2,i^*}\tilde{\mathbf{z}}_{2,i}^\top] \end{bmatrix} \mathbf{A}^\top \mathbf{W}_K^\top$$
$$= \mathbf{W}_Q \mathbf{B} \begin{bmatrix} \mathbb{E}[\mathbf{z}_{1,i^*}\mathbf{z}_{1,i^*}^\top] & \mathbf{0} \\ \mathbf{0} & \mathbb{E}[\mathbf{z}_{2,i^*}\tilde{\mathbf{z}}_{2,i}^\top] \end{bmatrix} \mathbf{A}^\top \mathbf{W}_K^\top$$

where the $\mathbf{0}$s in the last equality follow from the independence of $\mathbf{z}_1$ and $\mathbf{z}_2$, and the fact that $\mathbb{E}[\mathbf{z}_1] = \mathbb{E}[\mathbf{z}_2] = \mathbf{0}$.

Similarly computing the expectation for the negative condition (pairs of $\mathbf{q}, \mathbf{k}$ where the latent variable $\mathbf{z}_1$ differs, while $\mathbf{z}_2$ is held constant, i.e., $\mathbf{z}_{2,i} = \tilde{\mathbf{z}}_2$) yields

$$\mathbb{E}[\mathbf{q}\mathbf{k}^\top|-] =$$
$$\mathbf{W}_Q \mathbf{B} \begin{bmatrix} \mathbb{E}[\mathbf{z}_{1,i^*}\mathbf{z}_{1,i\neq i^*}^\top] & \mathbf{0} \\ \mathbf{0} & \mathbb{E}[\mathbf{z}_{2,i^*}\tilde{\mathbf{z}}_{2,i}^\top] \end{bmatrix} \mathbf{A}^\top \mathbf{W}_K^\top$$

Now we are left with the contrastive covariance matrix:

$$\Delta\mathbf{C}_{(\mathbf{z}_1)} = \mathbf{C}^+_{(\mathbf{z}_1)} - \mathbf{C}^-_{(\mathbf{z}_1)}$$
$$= \mathbf{W}_Q \mathbf{B} \begin{bmatrix} \mathbb{E}[\mathbf{z}_{1,i^*}\,\mathbf{z}_{1,i^*}^\top] - \mathbb{E}[\mathbf{z}_{1,i^*}\mathbf{z}_{1,i\neq i^*}^\top] & \mathbf{0} \\ \mathbf{0} & \mathbf{0} \end{bmatrix} \mathbf{A}^\top \mathbf{W}_K^\top$$

Thus $\Delta\mathbf{C}_{(\mathbf{z}_1)}$ isolates the contribution of latent variable $\mathbf{z}_1$ to the query-key interaction. This can be repeated for $\mathbf{z}_2$ by defining positive and negative conditions based on $\mathbf{z}_2$, while holding $\mathbf{z}_1$ constant.

The ranks and subspaces of latent variables can then be recovered by performing SVD on $\Delta\mathbf{C}_{(\mathbf{z}_1)}$ and $\Delta\mathbf{C}_{(\mathbf{z}_2)}$ respectively:

$$\Delta\mathbf{C}_{(\mathbf{z}_1)} = \mathbf{U}_{(\mathbf{z}_1)}\boldsymbol{\Sigma}_{(\mathbf{z}_1)}\mathbf{V}_{(\mathbf{z}_1)}^\top,$$
$$\Delta\mathbf{C}_{(\mathbf{z}_2)} = \mathbf{U}_{(\mathbf{z}_2)}\boldsymbol{\Sigma}_{(\mathbf{z}_2)}\mathbf{V}_{(\mathbf{z}_2)}^\top$$

The rank of $\mathbf{z}_1$ (denoted $r_1$) can be estimated by counting the number of singular values that captures 99% of the squared Frobenius norm of $\Delta\mathbf{C}_{(\mathbf{z}_1)}$. The top-$r_1$ singular vectors $\mathbf{U}^{[:r_1]}_{(\mathbf{z}_1)}$ and $\mathbf{V}^{[:r_1]}_{(\mathbf{z}_1)}$ give bases in query and key space that encode $\mathbf{z}_1$ respectively.

## C. Related Work

Since the adoption of attention modules in neural NLP models (Bahdanau, 2014; Sukhbaatar et al., 2015), researchers have been interested in better understanding them.

Often, researchers use the attention patterns itself as an explanation for a neural network (Li et al., 2016; Xu et al., 2015; Lee et al., 2025). This practice is not without contention: for instance, Jain & Wallace (2019) claims that "attention is not explanation" by carefully studying the relationship between attention weights and model outputs, in which they find low correlation between attention weights and feature importance. On the other hand, Wei et al. (2022) refutes back, suggesting that under certain conditions, attention scores can provide meaningful interpretations.

While attention patterns themselves may provide insight for a neural network's behavior, this begs the question, "why did the model attend to this token?" A growing line of work thus studies the inner mechanisms of attention.

This has been approached via multiple angles.

Similar to our work, some researchers have studied the QK space of attention heads to understand why a certain token is attended to, both in language and vision models.

In language, many of such works leverage features learned from sparse autoencoders (SAEs). Kamath et al. (2025); Ge et al. (2024); Friedman et al. (2025) decompose activations into SAE features and study aligned features from the query and key positions. Alternatively, researchers have used features recovered via training linear probes in order to observe how features interact in QK space. Gurnee et al. (2026) studies the mechanisms underlying a character count task, in which the model implicitly decides to produce a new line character when an implicit character limit is reached. By training probes for line widths and character counts, they demonstrate the two features interact in QK space. Lastly, Wynrow & Sharkey (2024) learns a sparse mask in QK space in order to detect matching features in QK space. Unlike prior work, our method does not rely on features from trained sparse autoencoders or probes, nor any training in order to retrieve QK features.

In vision, Pan et al. (2024); Doshi et al. (2026) apply SVD on query-key interactions to find QK features, such as channels communicating positional or content information, while Li et al. (2025) study how vision models bind tokens belonging to the same entity via bilinear probes trained in QK space.

Attention is often viewed as a "communication channel" that allows the model to exchange information from one token to another (Elhage et al., 2021). Merullo et al. (2024) study attention heads that likely "talk" to one another by decomposing attention weights using SVD and searching for aligned singular vectors across heads. Ahmad et al. (2025) extends this to include additional components (e.g., MLPs) to show low-rank subspaces that can be viewed as a unit of a computational circuit. Barbero et al. (2025) study communication channels in the rotary positional encodings of attention heads. Perhaps most related to our work is that of Franco & Crovella (2025) which similarly look for low-rank structure in attention heads that is critical for upstream usage in a circuit (i.e., computational graph).

Note that many of the works described above entail decomposing model weights or activations. While sparse autoencoders have been a popular choice of decomposition, other unsupervised methods include Neighbor Distance Minimization (Huang & Hahn, 2025), which may be a suitable tool to decompose QK spaces as well.

Lastly, researchers have also studied attention via visualizing feature interactions. Early works often visualized attention patterns over individual inputs as bipartite graphs (Liu et al., 2018; Strobelt et al., 2018; Vig, 2019) or heatmaps (Aflalo et al., 2022; Hoover et al., 2020; Kovaleva et al., 2019; Nanda et al., 2023), while subsequent work visualized the joint embedding space of keys and queries using PCA or UMAP to uncover global patterns of attention (Yeh et al., 2023).

## D. Training Details for Toy Model

Table 1 provides the hyperparameters used for training the toy model described in Section 4. We train until validation loss does not improve for more than 5 validation checks, where validation is performed every 200 training batches.

*Table 1.* Hyperparameters used for training the toy model.

| Hyperparameter | Value |
|---|---|
| $d$ | 32 |
| $d_{\text{head}}$ | 8, 16 |
| Batch size | 256 |
| Learning rate | 0.0001 |
| Weight decay | 0.01 |
| Validation batches | 20 |
| Validation batch size | 512 |
| Validation patience | 5 |

## E. Review of Binding Mechanisms

Here we review binding mechanisms from prior work (Feng & Steinhardt, 2023; Dai et al., 2024; Prakash et al., 2025; Gur-Arieh et al., 2025).

As a running example, consider a set of prompts that contain multiple pairs of entities that are grouped together (e.g., boxes containing objects), followed by a query regarding one of the entities:

```
The apple is in Box O. The banana
is in Box Z...(omitted)...
Which box is the banana in?
Answer:  Box
```

Assume we have $n$-pairs of entity and box pairs. We refer to each pair as an *entity group*, denoted as $(e_g, b_g)$ with entity $e_g$ and box $b_g$ for $g = 1, \ldots, n$.

How does the model answer this prompt? To our knowledge, Feng & Steinhardt (2023) is the first to suggest that models use "binding IDs": entities belonging in the same group are "tagged" with the same "binding ID", which the model uses to associate the two entities when queried later.

Prakash et al. (2025); Dai et al. (2024) further study similar settings and suggest that models assign "order-IDs" to entity groups based on their positions: the first entity group gets assigned the first order ID, while the second group gets assign a second order ID, and so on. When queried about an entity, the model retrieves the entity group associated with the corresponding order ID.

Finally, Gur-Arieh et al. (2025) show that order IDs are not the only "tags" used by models: they can also deploy "lexical" and "reflexive" tags to bind entities belonging to the same group. To summarize, we outline these three mechanisms of binding below:

**Order-ID (positional) mechanism.** The positional mechanism retrieves the answer based on the *group index $g$*. When queried about an entity $e_{g^*}$, the model uses the group index $g^*$ (e.g., "the third group") to fetch the corresponding box $b_{g^*}$. Put differently, it assumes an intermediate variable $Z_{\text{pos}}$ that encodes $g^*$ and retrieves the box associated with that index, regardless of the actual entity:

$$\text{Order-ID:} \quad Z_{\text{pos}} = g^* \quad \Rightarrow \quad \hat{b} = b_{Z_{\text{pos}}}.$$

where $\hat{b}$ is the retrieved box token.

**Lexical mechanism.** The lexical mechanism retrieves the answer by using the *identity of the queried entity*. This is perhaps the most intuitive, "correct" mechanism. When queried about an entity $e_{g^*}$, it assumes an intermediate vari-

able $Z_{\text{lex}}$ that encodes the entity identity, and retrieves the box from the group whose entity matches this identity:

$$\text{Lexical:} \quad Z_{\text{lex}} = e_{g^*} \quad \Rightarrow \quad \hat{b} = b_g \text{ such that } e_g = Z_{\text{lex}}.$$

**Reflexive mechanism.** The reflexive mechanism retrieves the entity group based on the *target box itself*. Informally, it assumes an intermediate variable $Z_{\text{ref}}$ that encodes the target box, suggesting that the model has already solved the query in an earlier computation step.

$$\text{Reflexive:} \quad Z_{\text{ref}} = b_{g^*} \quad \Rightarrow \quad \hat{b} = b_g \text{ such that } b_g = Z_{\text{ref}}.$$

## F. Additional Results

Here we provide additional results.

### F.1. Additional Results on Toy Model

Figure 11 shows the groundtruth ranks versus the recovered ranks from our method on additional models and tasks.

Figures 14 and 15 show results from causal interventions on additional models. Note that as the ranks of the two latent variables reach the number of attention head dimensions ($r_1 + r_2 \approx d_{\text{head}}$), the performance of the random baseline increases because at that point we are completely swapping out $\mathbf{k}_{i_{orig}}$ for $\mathbf{k}_{i_{target}}$.

Figure 13 show the interactions between the two latent variables in QK space when trained on our first task variant, i.e., discrete latent variables. Interestingly, unlike the continuous case, we no longer see symmetry in interactions.

### F.2. Additional Results on Semantic Categories and Binding Features

Figure 16 shows the effective ranks of $\Delta\mathbf{C}_{\text{order}}$, $\Delta\mathbf{C}_{\text{Lex.}}$ versus the number of entities used in constructing $\Delta\mathbf{C}$. While we see each head using different numbers of ranks, we see the effective ranks plateau after enough entities.

Figure 17 shows the PCA of keys and queries without projecting to our recovered order-ID and lexical subspaces. This reveals that order-ID is embedded in the first few principal components (PCs). While order-ID happens to have rank $\leq 3$ and thus can be captured with the first 3 principal components, PCA alone is unable to tell us the rank of QK features. Furthermore, PCA alone cannot inform us where other features (lexical) are encoded, unless one enumerates through all possible PC combinations.

Figure 18 shows causal intervention results on additional attention heads that attend to the correct binding entity (see Section 5.2).

## G. Qwen3-4B Results

Figure 19 provides causal interventions on Filter Heads of Qwen 3-4B-Instruct. Figure 20 provides causal intervention results for binding features.

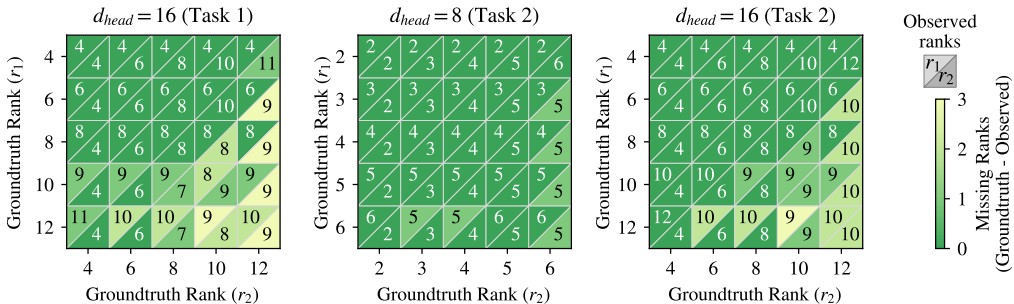

*Figure 11.* **Contrastive QK decomposition recovers the expected rank of each latent variable**, as long as there is no superposition (i.e., $r_1 + r_2 \leq d_{\text{head}}$). Each cell annotates the recovered ranks $r_1, r_2$, while the x and y-axes indicate the expected ranks. The color of each cell indicates the difference between expected and recovered ranks.

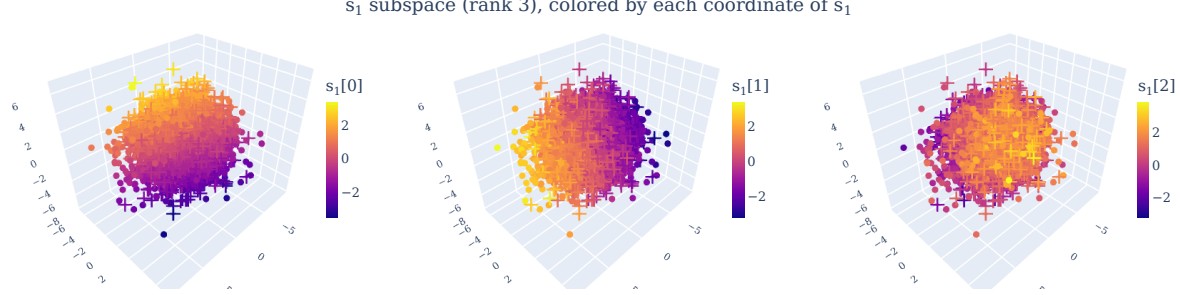

*Figure 12.* **PCA of Latent Variable Subspace (Second Task Variant).** The second toy task variant uses Gaussian hyperspheres as latent keys $\mathbf{s}_1, \mathbf{s}_2$, which is recovered by our method.

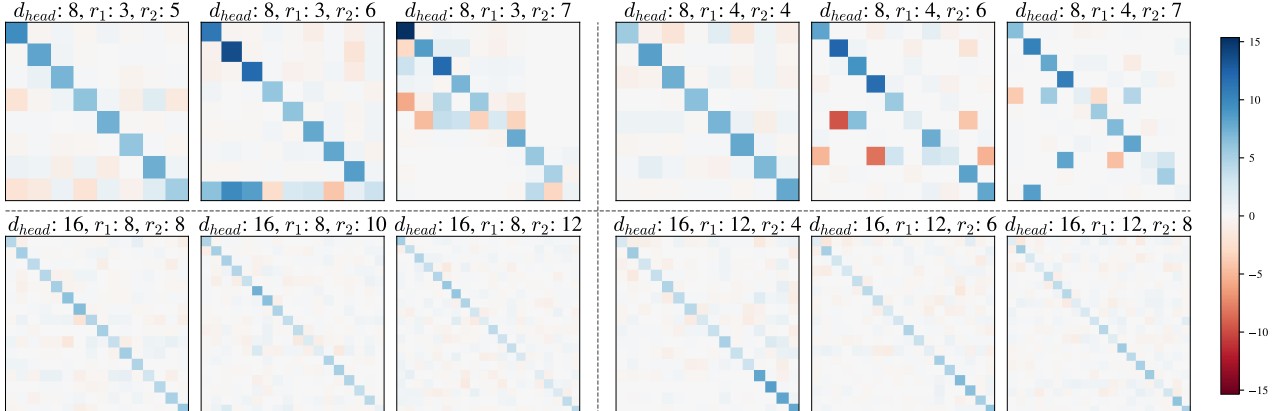

*Figure 13.* **Interactions between latent variables in QK space for models trained on discrete latent variables.** Interestingly, note that unlike the task with continuous latent variables (Figure 5), we do not see symmetric interactions in this case.

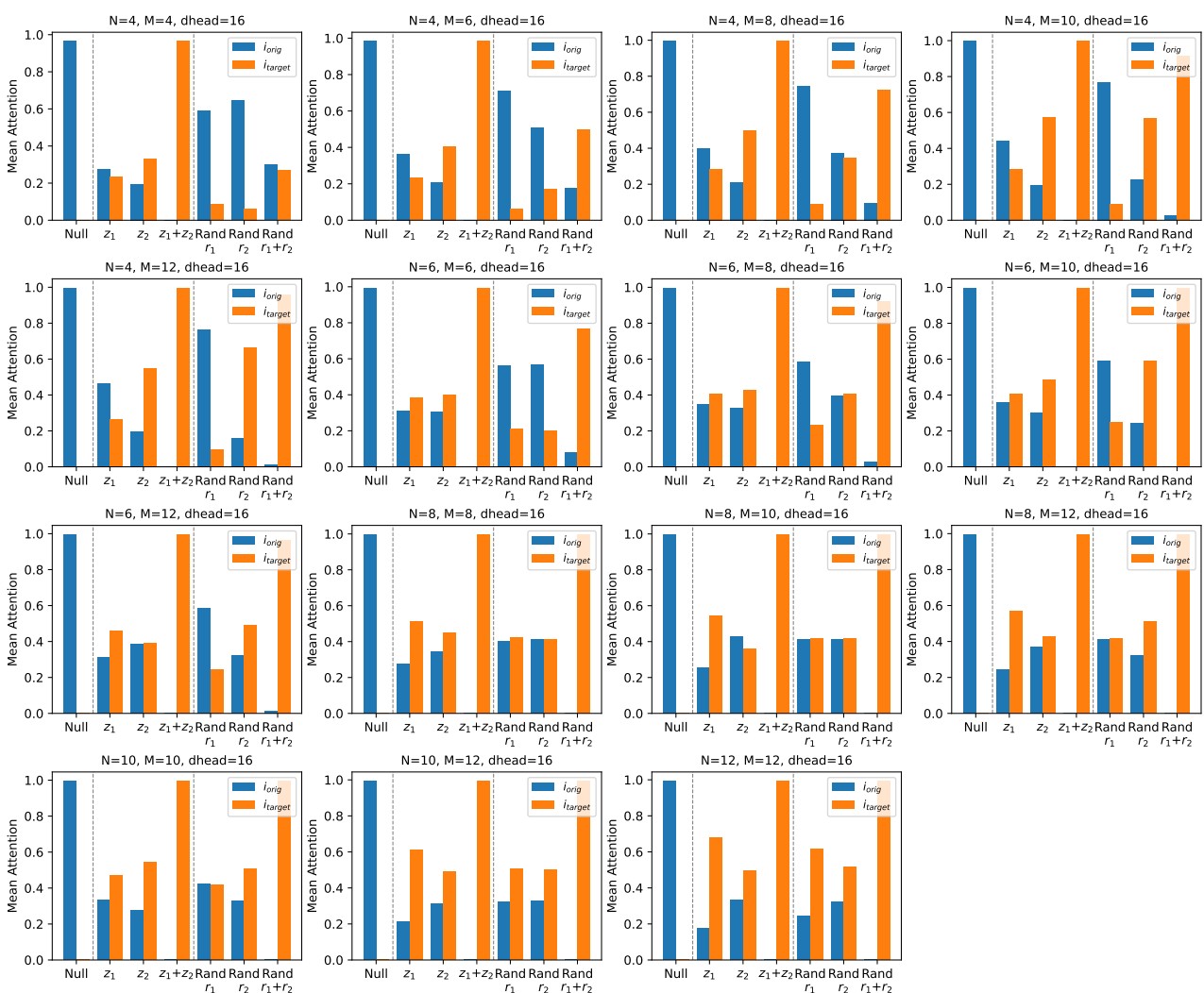

*Figure 14.* **Additional results for causal interventions on our toy model, for attention head with** $d_{\text{head}} = 16$**.**

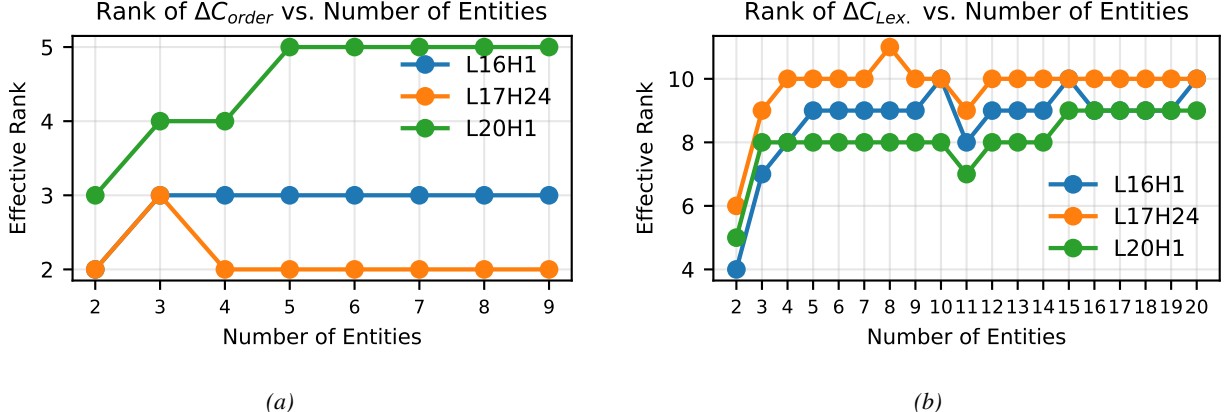

*Figure 15.* **Additional results for causal interventions on our toy model, for attention head with $d_{\text{head}} = 8$.**

*(a)*                                                                                    *(b)*

*Figure 16.* **Effective ranks vs. number of entities used in constructing $\Delta\mathbf{C}$.** While each head uses a different number of ranks, the effective ranks plateau after enough entities.

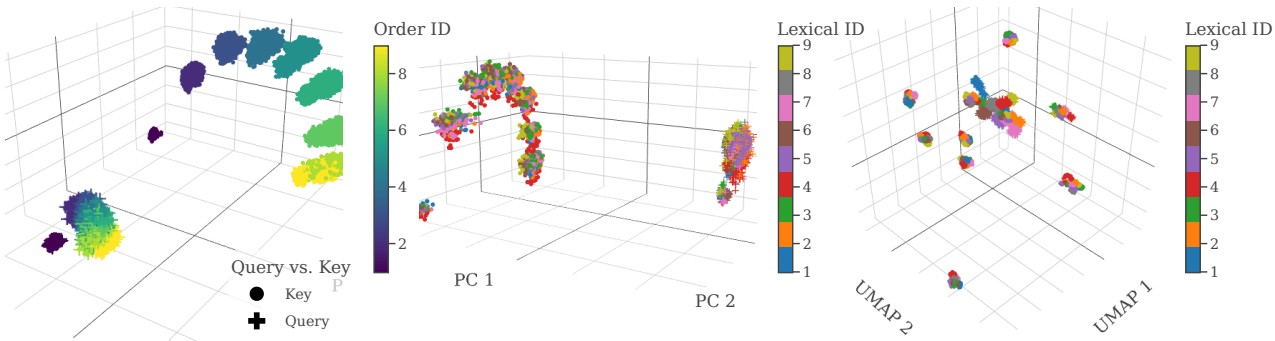

*Figure 17.* **PCA of keys and queries directly, before projecting onto our recovered QK subspaces.** Applying PCA on the keys and queries reveals that order-ID is encoded in the first few principal components (PCs). While order-ID happens to have rank $\leq 3$ and thus can be captured with the first 3 principal components, PCA alone is unable to tell us the rank of QK features. Furthermore, PCA does localize where other features (e.g., lexical) are encoded.

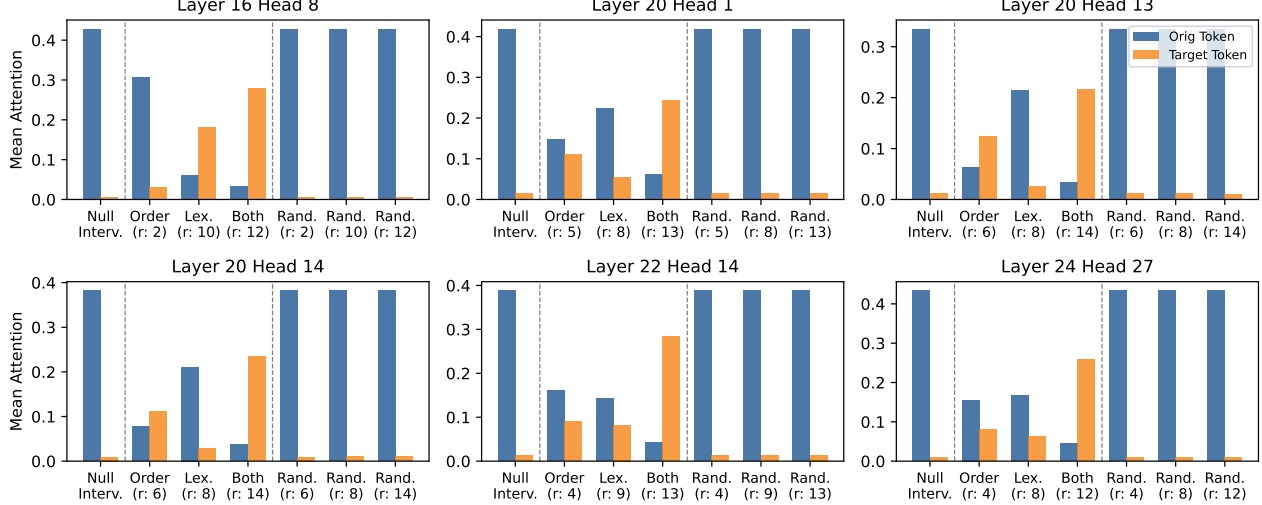

*Figure 18.* **Causal intervention results on additional binding heads.**

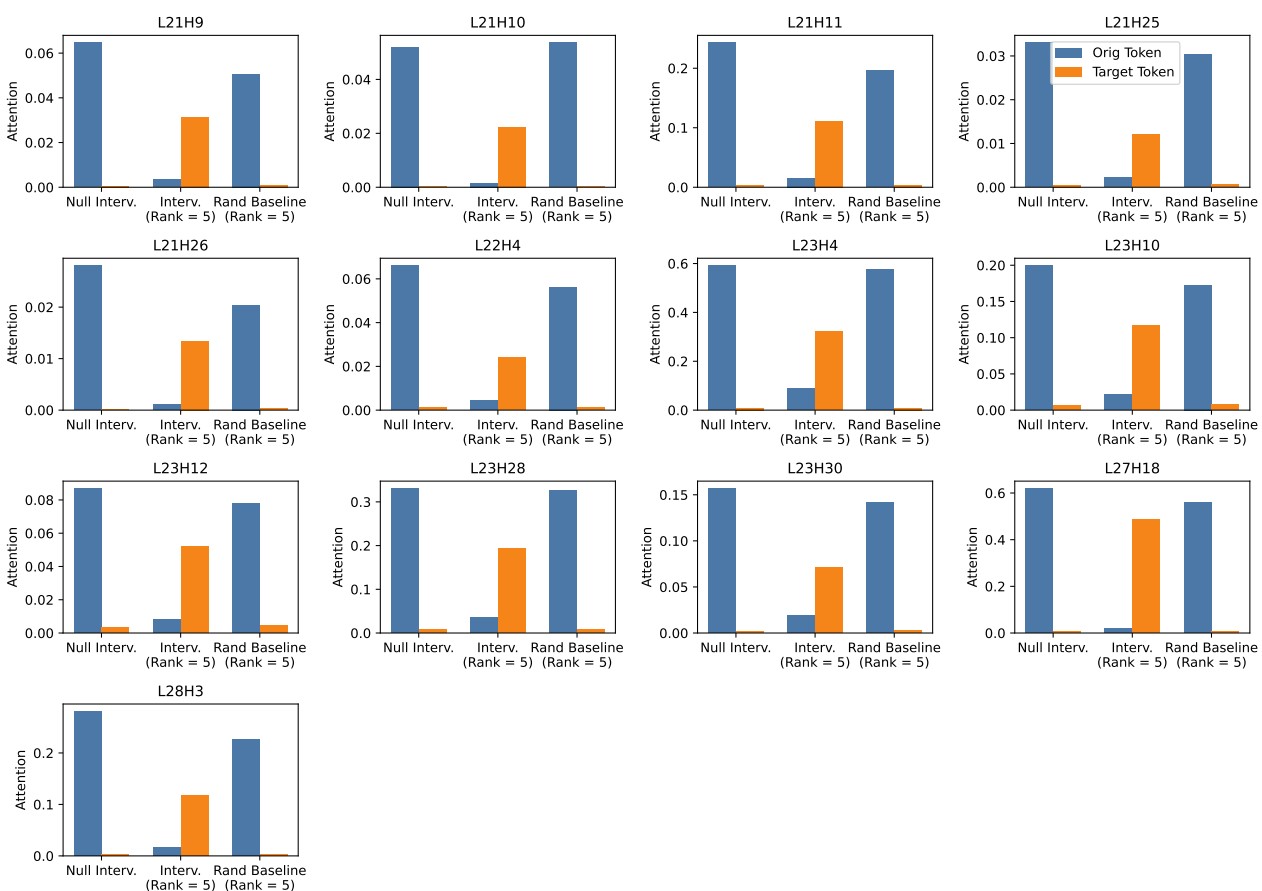

*Figure 19.* **Causal intervention results for Filter Heads on Qwen3-4B.**

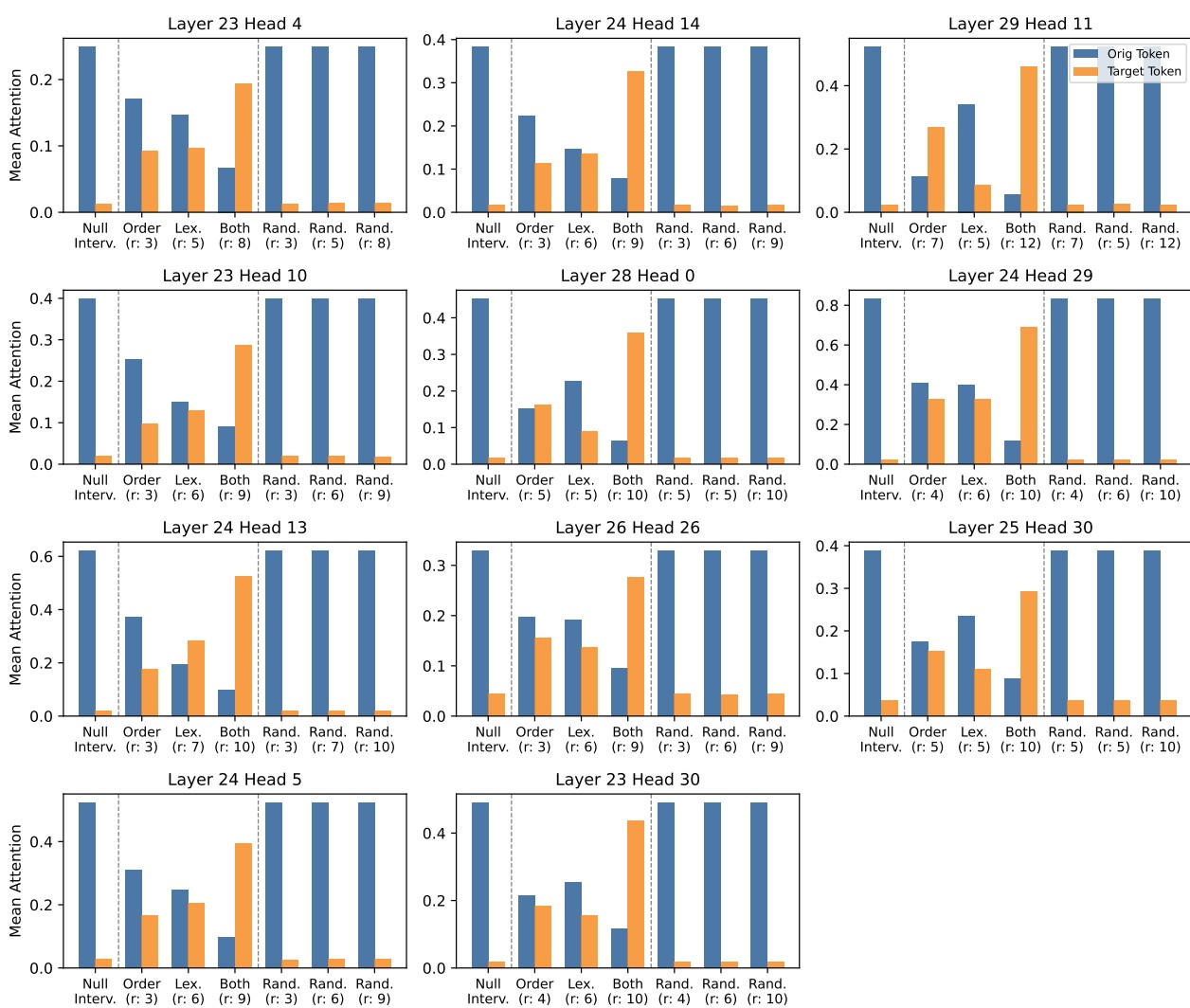

*Figure 20.* **Causal intervention results for binding on Qwen3-4B.**

