# OpenReview forum: "Decomposing Query-Key Feature Interactions Using Contrastive Covariances"
_ICML.cc/2026/Conference — ICML 2026 regular_

### Official Review · Reviewer_hXQ3 · 2026-02-25

**Soundness:** 3
**Presentation:** 3
**Significance:** 3
**Originality:** 3
**Overall Recommendation:** 4
**Confidence:** 2

**Summary:**

This paper introduces a mechanistic interpretability tool named Contrastive Covariance designed to understand the working mechanisms of attention heads by decomposing the Query-Key (QK) space—the bilinear joint embedding space between queries and keys. The core idea is to define a contrastive covariance term as the difference between "positive samples" (feature matches) and "negative samples" (feature mismatches) to isolate and extract low-rank QK subspaces associated with specific features. The authors first provide analytical and empirical validation using a toy model with known latent variables (both discrete and continuous), proving that the method accurately recovers the rank and subspaces of features. Subsequently, the method is applied to large language models (such as Llama 3.1 and Qwen 2.5), successfully identifying interpretable QK subspaces for categorical semantic features and binding features. The effectiveness of these subspaces is further validated through causal intervention experiments.

**Compliance With Llm Reviewing Policy:**

Affirmed.

**Final Justification:**

The authors addressed my concerns, so I would like to increase my score from 3 to 4.

**Key Questions For Authors:**

See weakness.

**Strengths And Weaknesses:**

Strengths:

- The paper not only proposes an intuitive contrastive covariance formula but also provides detailed analytical derivations in a toy model, clarifying the boundary conditions for recovering the rank and subspace of latent features (e.g., $r_1 + r_2 < d_{head}$).

- The method does not rely on complex optimization processes; it extracts feature directions simply via SVD from the covariance matrix. It is applicable across different architectures (Llama, Qwen) and various feature types (semantic, positional/binding).

- Through coordinate-replacement experiments (Causal Intervention) within the identified subspaces, the authors demonstrate that the extracted directions actually control changes in attention allocation, rather than merely being correlated with them.

- The experiments reveal phenomena such as "feature splits" and "superposition," pointing out potential discrepancies between "human-defined features" and "model-actual representation units," which provides profound insights for mechanistic interpretability research.

Weaknesses:

- The effectiveness of the method highly depends on how the contrastive conditions are defined. In complex tasks, if a single variable cannot be accurately isolated (i.e., holding all other factors constant), the contrastive covariance might capture mixed features or noise.

- Experiments show that when the model dimension is insufficient (superposition occurs), the extracted feature rank shifts and off-diagonal interference appears. This suggests the method's efficacy may be limited when analyzing highly compressed layers or models.

- While the method performs well in "filter heads," the paper notes that intervention experiments cannot transfer all attention scores, indicating that other significant features in the QK space remain uncaptured by this method.

---

> ### Author Rebuttal · Authors · 2026-03-29
>
> Thank you for your time and careful review!
>
> W1: > The effectiveness of the method highly depends on how the contrastive conditions are defined.
>
> Indeed, our method relies on carefully designed contrastive conditions. However, this point applies to most if not all supervised interpretability techniques, such as steering vectors and probes. All of these methods require contrastive labels (i.e., positive vs. negative) of a human-defined feature. Such datasets also need a careful design in order to avoid spurious correlations (i.e., mixed features or noise) from being recovered and mis-interpreted.
>
> These points are not unique to our method and are discussed in detail in prior work (ex: Section 2.2.2 “Probes need carefully chosen data for well-defined concepts” of [1], or Section 4.4 “Datasets vs. Tasks” and Section 4.5 “Properties Must be Pre-defined” of [2]).
>
> Despite these challenges, our field has made meaningful progress in interpreting LLMs. For instance, supervised probes and steering vectors pointed to the possibility of linear feature representations, which has led to unsupervised methods like sparse auto-encoders (SAEs) for feature discovery, which then improved circuit discoveries. In a similar manner, we believe that our findings (e.g., low-rank feature representations) update our understanding of feature representations and anticipate that they will lead to new meaningful research directions.
>
>
> [1] Sharkey et al. Open Problems in Mechanistic Interpretability. TMLR 2025 https://openreview.net/pdf?id=91H76m9Z94
> [2] Belinkov. Probing Classifiers: Promises, Shortcomings, and Advances. CL 2022 https://aclanthology.org/2022.cl-1.7/
>
>
> W2: > …the method’s efficacy may be limited when analyzing [superposition].
>
> As you point out, we view the discovery of superposition in QK space of toy models a “profound insight”.
> While we dedicate half a page to explicitly discuss its implications for our method, in the LLM heads we study, the feature representations we find are low-rank, stable across an increasing number of entities, with strong causal effects. These subspaces even generalize across languages and out-of-domain test cases, where the subspaces derived from one language still demonstrate robust causal effects on prompts of another language – please see our response to reviewer dDYc. So while it is yet unclear to what extent superposition may occur in QK space (which we view as an interesting question for our community!), for the particular heads and domains we analyze, our method still recovers useful and interpretable structure.
>
>
> W3: > While the method performs well in "filter heads," … other significant features in the QK space remain uncaptured by this method.
>
> Yes, our findings suggest that additional features are being used by the attention heads in addition to the categorical features that our method recovers. However, our method is not designed to discover all features. Rather, our method is designed to recover the exact subspace in which a feature of interest is encoded in.
>
> Revisiting the analogy to steering vectors vs. SAEs, a steering vector might find a 1-dimensional subspace for a feature with meaningful causal effects, although we do not expect it to be the only meaningful feature being used by the model. Meanwhile, unsupervised SAEs might discover a more exhaustive bag of potential features, although these features may be less “faithful” (given issues like identifiability [3], feature splits/absorptions [4], or sometimes lacking causal effects [5]).
>
> In a similar manner, our contrastive covariance method is meant to recover the precise subspace in which features are encoded, which is a separate problem from feature discovery. With that being said, we do not view your point (additional features existing outside of the categorical features we identify) as a weakness, as this is not something our method is designed to address.
>
>
>
> [3] Paulo et al. Sparse Autoencoders Trained on the Same Data Learn Different Features. https://arxiv.org/abs/2501.16615
> [4] Chanin et al. A is for Absorption: Studying Feature Splitting and Absorption in Sparse Autoencoders. https://arxiv.org/abs/2409.14507
> [5] Arad et al. SAEs Are Good for Steering – If You Select the Right Features. https://arxiv.org/pdf/2505.20063
>
>
> –
>
> Thank you again for your time and feedback! We are happy to engage in additional questions or feedback if any.

---

> > ### Author Rebuttal · Reviewer_hXQ3 · 2026-04-01
> >
> > Thanks for the author's response. After reading the rebuttal and reviews by other reviewers, my concerns are addressed. I would like to increase my rating.

---

### Official Review · Reviewer_M86Q · 2026-03-10

**Soundness:** 4
**Presentation:** 4
**Significance:** 3
**Originality:** 3
**Overall Recommendation:** 5
**Confidence:** 4

**Summary:**

This paper presents a method to decompose the QK computation of the attention mechanism using contrastive covariances. Given a feature of interest (e.g., fruits), the idea is to use positive and negative samples to construct positive and negative covariances of the $qk$ outer product, which captures the interaction between queries and keys. Then, the difference of these covariances is the object of study of the paper. The method is validated in two scenarios: a toy setting with known latent variables and in real models. In the toy setting, the model recovers the rank of the latent variables when there is no superposition, and the latent variables can be recovered using PCA. In real models, the method recovers the subspaces encoding categories in Filter Heads, and also recovers the subspaces responsible for order ID and lexical information in the binding features task. This result was also used to decompose the entire attention logit computation (given a query token and all possible key tokens) as a sum of order, lexical, and other terms. Moreover, all the claims are supported with causal intervention results.

**Compliance With Llm Reviewing Policy:**

Affirmed.

**Final Justification:**

My evaluation haven't changed with the rebuttal, and I think that this is a solid paper. So, I kept my original score.

**Key Questions For Authors:**

No questions for the authors.

**Limitations:**

Yes.

**Strengths And Weaknesses:**

## Strengths:

- The method proposes a supervised method to find the QK features. This is novel in comparison to most of the literature, which leverages unsupervised methods such as SVD to study the QK.
- The paper is very well written and easy to read.
- All the claims are followed by a causal intervention experiment, which makes the claims stronger.
- Finding and understanding the features used by an attention head is very relevant for interpretability and it can have multiple applications, such as circuit tracing.
- The paper is very transparent about the weakness of the method.

## Weaknesses:

1. The contrastive objective may limit the kind of features that this method can identify. Features that appear in both sets (negative and positive) may be cancel out.
2. This method also requires a very well designed supervised dataset and a human defined feature of interest. As the authors mentioned, this may not reflect the way that models actually represent features internally.

---

> ### Author Rebuttal · Authors · 2026-03-29
>
> Thank you for your time and careful review!
>
> W1: You are certainly right that features that appear both in the positive and negative covariance terms would cancel out.
> However, this is precisely what we want: if an attention head relies on more than one feature, we want to factor out and isolate each one by canceling out the others. Admittedly this requires a very careful design of the contrastive data, which we discuss further below.
>
> W2: You are also correct in that our method requires a supervised dataset and a human-defined notion of a feature beforehand. However, this point applies to most if not all supervised interpretability techniques, such as steering vectors and probes. All of these methods require labels (i.e., positive vs. negative) of a human-defined feature. Such datasets also need careful designs in order to avoid spurious correlations being recovered.
>
> These points are not unique to our method and are discussed in detail in prior work (ex: Section 2.2.2 “Probes need carefully chosen data for well-defined concepts” of [1], or Section 4.4 “Datasets vs. Tasks” and Section 4.5 “Properties Must be Pre-defined” of [2]).
>
> Despite these challenges, our field has made meaningful progress in interpreting LLMs. For instance, supervised probes and steering vectors pointed to the possibility of features being linearly represented, which has led to unsupervised methods like sparse auto-encoders, which then improved circuit discoveries. In a similar manner, we believe that our findings (e.g., low-rank feature representations) update our understanding of feature representations and anticipate that they will lead to new meaningful research directions.
>
>
> Thank you again for your time and feedback!
>
> [1] Sharkey et al. Open Problems in Mechanistic Interpretability. TMLR 2025 https://openreview.net/pdf?id=91H76m9Z94
> [2] Belinkov. Probing Classifiers: Promises, Shortcomings, and Advances. CL 2022 https://aclanthology.org/2022.cl-1.7/

---

> > ### Author Rebuttal · Reviewer_M86Q · 2026-03-31
> >
> > I had no major concerns about the paper. It is a solid paper in my opinion.

---

### Official Review · Reviewer_hGtb · 2026-03-12

**Soundness:** 2
**Presentation:** 1
**Significance:** 2
**Originality:** 3
**Overall Recommendation:** 3
**Confidence:** 2

**Summary:**

To understand the Query-Key space, the paper proposes to use Contrastive Covariances so that the space is decomposed into low-rand and human interpretable space. The authors tried experiments with a simple setting and large language models. Based on the decomposed space, they show how the attention scores can be interpreted.

**Compliance With Llm Reviewing Policy:**

Affirmed.

**Final Justification:**

Some of my initial concerns have been resolved.

**Key Questions For Authors:**

1. Could the authors more explicitly state the primary contributions compared to the prior work? What kind of insight will readers gain by reading this paper?

2. Please provide the meaning of the tasks (variants 1 and 2)? It would be helpful to see a clear explanation on how they are connected to QK decomposition?

3. Please specify which layer from the Llama 3.1-8B Instruct model were analyzed?

4. For the sake of reproducibility, how can we have the tokens belonging (and not belonging) to the queried category in Section 5.1?

5. The manuscript would be strengthened by interpreting the experiment results instead of simply describing or observing them. For example, about "the structures of queries and keys in PC 2 and 3 are symmetric to one another" and "clear semantic clusters", what do they mean?

6. I don't understand how to replace "the QK components of tokens from one category (e.g., fruits) with those from another category (e.g., animals)." in Figure 7. Could you elaborate it more?

7. It was a bit hard for me to understand the paper. A more logical reorganization and more precise language would significantly enhance its accessibility.

**Limitations:**

See the Key Questions for Authors above.

**Strengths And Weaknesses:**

It is crucial to understand the Query-Key space in the attention mechanism especially in the Transformer architecture. The paper tries a new approach (Contrastive Covariance) to understand the Q-K space.

However, while the paper addresses the important topic of architectural biases in Transformers, several weaknesses limit its impact. First, the primary contribution relative to prior work remains unclear, and the manuscript would benefit from a more explicit statement of the novel insights it offers. Second, methodological ambiguity hinders reproducibility. Third, the paper tends to describe experimental observations without providing a deeper interpretation of their implications. Finally, the overall accessibility of the research is hampered by a lack of logical organization and a need to clarify complex concepts.

---

> ### Author Rebuttal · Authors · 2026-03-29
>
> Thank you for your time and careful review!
>
> 1. Thank you for this feedback, we will incorporate this for our next revision. While we believe our work provides a few contributions, here we briefly discuss what we view as the most important contributions compared to prior work. Namely, our method allows one to identify human-interpretable low-rank subspaces of features in the QK space of models. This differs from prior work in two important ways: first, prior work that attempts to interpret attention relies on pre-existing features (e.g., sparse autoencoders) or other extra optimizations (e.g., training probes, masks, etc.). Our method is able to extract feature subspaces without either pre-requisites. These points are discussed in Section 6.
>
> The other perhaps more important difference is that these previous methods rely on an implicit assumption of rank-1 feature representations. Our work provides evidence for low-rank feature representations rather than simple rank-1 representations, and we believe future interpretability methods will benefit with this updated perspective of how features are represented.These points are discussed in Section 7.
>
> 2. Our key takeaway is that latent features in QK space are encoded in different subspaces of ranging ranks. Our toy task is designed to reflect this insight very closely, where we randomly sample latent variables from different distributions (which correspond to the two different task variants). The first task variant samples latent variables from {+/- 1}^r, where r is the rank of the latent variable, whereas the second task variant samples from Gaussian distributions of different ranks. These latent variables are then embedded into a common embedding space, emulating our mental model of how attention heads work.
>
> 3. We study attention heads that exhibit specific behaviors (Filter Heads, binding). The criteria for identifying these heads are specified in the text (lines 260-266 for Filter Heads, lines 316 to 324 for binding heads). For Filter Heads, this results in heads [(Layer 16, Head 19), (Layer 20, Head 14), (Layer 20, Head 26)]. Figures 6 and 7 labels results for each head. For binding heads, this results in [(Layer 16, Head 1), (Layer 16, Head 19), (Layer 17, Head 24)]. Figure 9 labels results for each head. Figure 8 corresponds to Layer 16 Head 1, although similar results can be found in all of our identified heads.
>
> 4. Our supplementary code submission (and eventual open-source repo) has all the tokens used for each category in Section 5.1.
>
> 5. If I understand correctly, I believe this point is regarding the PCA visualizations - every visualization in our paper is coupled with causal intervention experiments to confirm our interpretations (Figure 3 goes with Figure 4, Figure 6 goes with Figure 7, Figure 8 goes with Figure 9).
>
> The “clear semantic clusters” suggest that the coordinates in each of our identified subspaces correspond to an interpretable feature, or put differently, that we have found the precise subspaces in which a feature of interest is encoded. The “structures of queries and keys are symmetric to one another” mean that when features in query and key spaces are in alignment, high attention scores are produced.
>
> Again, these statements are backed by causal interventions for all of our PCA plots. Our causal experiments are designed to confirm these statements, which work by swapping the coordinates of key vectors within these feature subspaces from one cluster to another, in order to control the alignment between keys and queries and thus shift the attention from one token to another.
>
>
> 6. All causal intervention experiments use the same method, which are described in Equations 8, 9, and 10. In particular, given the key vectors corresponding to fruit tokens and key vectors corresponding to animal tokens, we project each of them onto the categorical semantic subspace as described in Section 5.1. Then the coordinates of key vectors within these subspaces are swapped between that of fruits and animals, after which the forward pass is resumed, allowing us to measure the change in attention scores before and after our interventions.
>
> 7. Thank you for this feedback. We are happy to incorporate suggestions regarding re-organization or more precise language if you have concrete suggestions or specific parts that were unclear.

---

> > ### Author Rebuttal · Reviewer_hGtb · 2026-04-06
> >
> > I would like to thank the authors for their detailed responses to my comments. I will increase my score accordingly. However, my concerns regarding the paper’s overall contribution—specifically, how these findings provide significant value to other researchers in the field—have not been fully addressed.

---

> > > ### Author Response · Authors · 2026-04-06
> > >
> > > Thank you for your response! We believe our response to question 1 above answers your question, but here we elaborate how our contributions provide value to other researchers in our field.
> > >
> > > Namely, in the past few years, interpretability researchers have mostly assumed a "Linear Representation Hypothesis" -- that interpretable features are represented as simple **rank-1 "directions"** in activation space [1, 2, 3]. Broadly, this led to two lines of work:
> > >
> > > 1) An abundance of work studied and found interpretable rank-1 features in language models, which people often refer to as "steering vectors" [4, 5, 6, 7, 8].
> > >
> > > 2) By characterizing activations as a sum of sparse linear directions, numerous researchers developed various sparse auto-encoders (SAEs), which has arguably become the prevailing method for interpreting LLMs [9, 10, 11, 12, 13].
> > >
> > > Importantly, our work provides researchers a significant update as to how features are encoded -- we find evidence that attention heads use low-rank features (as opposed to rank-1 directional features) to decide which token to attend to. This is an important update as to how our community should think about interpreting large language models -- as opposed to a bag of linear feature directions, we should consider them as a mix of low-rank feature subspaces, in which the coordinates in each subspace represent a feature. With this being said, we believe future SAE architectures should be designed with this insight in mind. Note that we make this call to action to the community in our final concluding paragraph.
> > >
> > >
> > > Apart from informing researchers about the geometry of feature representations, we also fill a gap in our understanding of language models: despite attention being at the heart of Transformers, our community still struggles to interpret why an attention head attends to the tokens it decides to attend to. We take a step towards filling this gap. Note that prior efforts still relied on an incomplete assumption of the rank-1 features discussed above. As already discussed in our original response to question 1 above (as well as Section 6 and Section C), our work further differs from prior work in that we develop a principled and simple approach that does not assume rank-1 features, does not require any optimization, and does not require any pre-existing features (SAE features).
> > >
> > > To summarize, as reviewer M86Q notes, "Finding and understanding the features used by an attention head is very relevant for interpretability and it can have multiple applications, such as circuit tracing."
> > >
> > > We hope this answers your question. If you have any specific comments, questions, or concerns, please be specific and let us know and we are happy to discuss. If there are no remaining questions or concerns, we ask the reviewer to consider raising their score.
> > >
> > > Thank you!
> > >
> > > [1] Park et al. "The Linear Representation Hypothesis and the Geometry of Large Language Models". 2024
> > > [2] Nanda et al. "Emergent linear representations in world models of self-supervised sequence models. 2023.
> > > [3] Jiang et al. "On the Origins of Linear Representations in Large Language Models". 2024
> > >
> > > [4] Panickssery et al. "Steering Llama 2 via Contrastive Activation Addition". 2023
> > > [5] Zou et al. "Representation Engineering: A Top-Down Approach to AI Transparency", 2023
> > > [6] Chen et al. "Designing a Dashboard for Transparency and Control of Conversational AI", 2024
> > > [7] Lee et al. "A Mechanistic Understanding of Alignment Algorithms: A Case Study on DPO and Toxicity". 2024
> > > [8] Park et al. "The Geometry of Categorical and Hierarchical Concepts in Large Language Models". 2025
> > >
> > > [9] Bricken et al. "Towards Monosemanticity: Decomposing Language Models With Dictionary Learning". 2023
> > > [10] Cunningham et al.l "Sparse Autoencoders Find Highly Interpretable Features in Language Models". 2023
> > > [11] Wu et al. "Scaling and evaluating sparse autoencoders". 2025
> > > [12] Shu et al. "A Survey on Sparse Autoencoders: Interpreting the Internal Mechanisms of Large Language Models". 2025
> > > [13] Dunefsky et al. "Transcoders Find Interpretable LLM Feature Circuits". 2024

---

### Official Review · Reviewer_dDYc · 2026-03-12

**Soundness:** 4
**Presentation:** 4
**Significance:** 3
**Originality:** 3
**Overall Recommendation:** 5
**Confidence:** 4

**Summary:**

The paper proposes a contrastive covariance method to analyze why an attention head attends to particular tokens by decomposing the query–key (QK) interaction space into low-rank, human-interpretable components. The core idea is to construct positive vs. negative query–key covariance matrices for a feature of interest, while holding other factors fixed, and then take their difference to isolate a feature-specific QK subspace.
The authors validate the method in a controlled toy “payload retrieval” task with known latent variables, showing that the decomposition can recover latent ranks and subspaces when capacity permits, and they use causal interventions to demonstrate that manipulating the recovered subspaces predictably shifts attention.
Finally, they apply the approach to LLMs to identify low-rank QK subspaces for (1) categorical “filter head” behavior over lists and (2) binding mechanisms and they propose a straightforward attention-logit attribution by projecting queries onto discovered feature subspaces and decomposing logits into per-feature contributions plus residual.

**Compliance With Llm Reviewing Policy:**

Affirmed.

**Key Questions For Authors:**

1- How sensitive are the recovered subspaces to the exact definition of the positive/negative contrast?

2- when you swap or modify components of keys/queries, how do you ensure you’re not introducing out-of-distribution artifacts that change behavior for unrelated reasons?

3- How robust are the LLM case studies across prompts and contexts? If you change template, list length, distractors, languages, or tokenization quirks, do the same subspaces appear?

**Limitations:**

yes

**Strengths And Weaknesses:**

Strengths:
- The method is conceptually lightweight (contrastive covariance + SVD) and does not require training auxiliary models, which is a practical advantage.
- The paper not only claims subspace recovery but also tests it with causal interventions that shift attention in the intended ways, plus it explicitly examines failure modes.
- The filter-head categorical subspaces (often rank-1 per category in their setup) and binding subspaces (lower-rank order-ID vs. higher-rank lexical) are interpretable, and tied to concrete prompt constructions and visualizations/interventions.

Weaknesses:
- The method depends on constructing positive/negative pairs that “isolate” a feature; this presupposes you already know what feature you’re looking for and that it can be cleanly contrast-controlled. The paper acknowledges this limitation and frames unsupervised discovery as future work, but as-is the method is not a general-purpose discovery tool.
- The toy analysis shows rank recovery can fail under superposition, and in LLMs feature subspaces may overlap. The later logit attribution is also order-dependent when subspaces aren’t distinct, which can make quantitative attributions sensitive.

---

> ### Author Rebuttal · Authors · 2026-03-29
>
> Thank you for your time and careful review!
>
> W1:
> Indeed, our method relies on contrastive conditions. However, this point applies to most if not all supervised interpretability techniques, such as steering vectors and probes. They all require contrastive labels (i.e., positive vs. negative) of a human-defined feature, and such datasets need careful designs to isolate specific features and avoid spurious correlations.
>
> These points are not unique to our method and are discussed in prior work (ex: Section 2.2.2 “Probes need carefully chosen data for well-defined concepts” of [1], or Section 4.4 “Datasets vs. Tasks” and Section 4.5 “Properties Must be Pre-defined” of [2]).
>
> To continue the analogy with probes, a supervised probe might identify an exact rank-1 subspace for a feature - but a probe is typically not designed to be a general-purpose discovery tool, in a similar manner in which our method is meant to identify the local subspace of a feature, and not for general-purpose feature discovery.
>
> Despite the challenges above, our field has made meaningful progress. Supervised probes suggested the possibility of linear representations, which led to unsupervised methods like SAEs for feature discovery, which then improved circuit discoveries. In a similar manner, we believe that our findings (e.g., low-rank feature subspaces in QK space) can lead to general-purpose discovery tools.
>
> [1] arxiv.org/abs/2501.16496
> [2] arxiv.org/abs/2102.12452
>
>
> W2:
> Perhaps the overarching point in these two comments is to what extent features are superimposed in QK space. This is an interesting & open question to pose to our community. While our toy analysis reveals a potential pitfall of our method (which we dedicate half a page to discuss), we view this as a meaningful insight rather than a weakness. This question also motivates our call for an unsupervised QK decomposition, analogous to SAEs as a potential tool for superposition in activations.
>
>
> Q1:
> They are indeed somewhat sensitive. As an example, when recovering the Order-ID subspace (rank 3), if the set of entities used in the dataset are not held fixed, the resulting subspace has rank 11, which is likely a mix of both the lexical and order-ID features.
>
> Q2:
> This is a great question! A few thoughts: first, in our LLM settings, note that in our random baselines, in which we swap the components in random subspaces of the same rank, the attention barely shifts, suggesting that any generic perturbation has no causal effect. This provides some evidence that noise (OOD artifacts) does not affect the attention heads in unexpected ways.
>
> Second, in principle, if one’s contrastive data is designed carefully enough, all other latent variables (including OOD artifacts) should get canceled out: see our analytical explanation in Section B of our appendix.
>
> Lastly, we have included an additional OOD test and observe robust OOD generalization of our recovered subspaces – see below.
>
>
> Q3:
>
> Regarding list length - with enough entities in each prompt, we see the ranks of each feature converge - see Figure 16 of our appendix.
>
> We also ran 3 new experiments:
> 1) We add distractors (i.e., random names) to our prompts. Ex: “Mary put the apple in Box B, John put the pencil in Box G... Which box is the pencil in?”
>
> 2) We use a new language (Korean)
>
> 3) We test for OOD generalization by deriving covariance terms using our original prompts, but test the recovered subspaces on Korean prompts.
>
> We report causal intervention results as they are likely the most important experiments.
>
> For all tasks, we see the same set of heads attend strongly to the correct token, and intervention results all look similar as before. For brevity we report the top head (L17 H24), though the remaining heads yield similar results and will be reported in our next revision.
>
> 1) Distractors
> |   | Mean Att. | Mean Att. |
> |---|:---:|:---:|
> |    | Orig. Token | Target Token |
> | Null Interv. | 0.70 | 0.01 |
> | Order-ID (rank 6) | 0.14 | 0.31 |
> | Lex. (rank 9)  | 0.17 | 0.23 |
> | Both (rank 15) | 0.02 | 0.78 |
> | Rand. Baseline (rank 6) | 0.63 | 0.01 |
> | Rand. Baseline (rank 9) | 0.61 | 0.02 |
> | Rand. Baseline (rank 15) | 0.55 | 0.03 |
>
> 2) Korean
>
> |   | Mean Att. | Mean Att. |
> |---|:---:|:---:|
> |   | Orig. Token | Target Token |
> | Null Interv. | 0.49 | 0.01 |
> | Order-ID (rank 3) | 0.28 | 0.09  |
> | Lex. (rank 7) | 0.17 | 0.12 |
> | Both (rank 10)  | 0.07 | 0.29 |
> | Rand. Baseline (rank 3) | 0.48 | 0.01 |
> | Rand. Baseline (rank 7) | 0.45  | 0.01 |
> | Rand. Baseline (rank 10) | 0.44 | 0.02 |
>
> 3) OOD
>
> |   | Mean Att. | Mean Att. |
> |---|:---:|:---:|
> |   | Orig. Token | Target Token |
> | Null Interv. | 0.57 | 0.01 |
> | Order-ID (rank 2) | 0.41 | 0.05  |
> | Lex. (rank 11) | 0.10 | 0.28 |
> | Both (rank 13)  | 0.06 | 0.38 |
> | Rand. Baseline (rank 2) | 0.55 | 0.01 |
> | Rand. Baseline (rank 11) | 0.48  | 0.01 |
> | Rand. Baseline (rank 13) | 0.45 | 0.02 |
>
> We believe these results strengthen our work, and thank the reviewer for their question!

---

> > ### Author Rebuttal · Reviewer_dDYc · 2026-04-01
> >
> > I thank the reviewers for addressing all my concerns.

---

### Decision · Program_Chairs · 2026-04-30

**Decision:**

Accept (regular)

**Comment:**

The paper address an interpretability question, i.e., why an attention head attends to a particular token, by proposing a simple contrastive-covariance decomposition of QK space. During rebuttal, the reviewers generally agreed the method's technical soundness and practical significance, and the authors addressed robustness concerns with additional empirical evidence. There are some remaining concerns that the method still requires carefully designed supervised contrasts and human-specified features. Furthermore, the method's broader generality under superposition and across more diverse heads/tasks also need further validated. Overall, I recommend Accept.